# CSF proteome profiling reveals biomarkers to discriminate dementia with Lewy bodies from Alzheimer´s disease

Marta del Campo [1,2,3] ✉, Lisa Vermunt[1,4], Carel F. W. Peeters[5], Anne Sieben[6], Yanaika S. Hok-A-Hin [1], Alberto Lleó[7,8], Daniel Alcolea [7,8], Mirrelijn van Nee[9], Sebastiaan Engelborghs [10,11,12], Juliette L. van Alphen[4], Sanaz Arezoumandan[13], Alice Chen-Plotkin[13], David J. Irwin[13], Wiesje M. van der Flier [4,9], Afina W. Lemstra[4] & Charlotte E. Teunissen [1]

Diagnosis of dementia with Lewy bodies (DLB) is challenging and specific biofluid biomarkers are highly needed. We employed proximity extension-based assays to measure 665 proteins in the cerebrospinal fluid (CSF) from patients with DLB ($n = 109$), Alzheimer´s disease (AD, $n = 235$) and cognitively unimpaired controls ($n = 190$). We identified over 50 CSF proteins dysregulated in DLB, enriched in myelination processes among others. The dopamine biosynthesis enzyme DDC was the strongest dysregulated protein, and could efficiently discriminate DLB from controls and AD (AUC:0.91 and 0.81 respectively). Classification modeling unveiled a 7-CSF biomarker panel that better discriminate DLB from AD (AUC:0.93). A custom multiplex panel for six of these markers (DDC, CRH, MMP-3, ABL1, MMP-10, THOP1) was developed and validated in independent cohorts, including an AD and DLB autopsy cohort. This DLB CSF proteome study identifies DLB-specific protein changes and translates these findings to a practicable biomarker panel that accurately identifies DLB patients, providing promising diagnostic and clinical trial testing opportunities.

Dementia with Lewy Bodies (DLB) is one of the most common forms of dementia in the aged population after Alzheimer's disease (AD)[1] and is clinically characterized by cognitive fluctuations, visual hallucinations, parkinsonism and rapid eye movement sleep behavior disorder. DLB is pathologically characterized by the intraneuronal accumulation of α-synuclein (α-syn) in Lewy bodies in the neocortex[2]. The clinical and pathological presentation strongly overlap with AD, challenging differential diagnosis and leading to a large proportion of miss- or undiagnosed DLB patients[3–5].

Limited number of biomarkers have been widely analyzed to date in DLB. Despite previous studies on α-syn in cerebrospinal fluid (CSF) showing conflicting results[6–8], recent developments using seed amplification assays allow to detect α-syn brain proteinopathy in CSF and skin[9–12]. These assays can discriminate DLB from control or AD patients

with high accuracy[9–12]. However, α-syn pathology is not unique for DLB patients and more than 40% of AD cases can present with this comorbid pathology[13]. Similarly, previous studies have shown that the core CSF biomarkers used to support AD diagnosis (amyloid β peptide (Aβ$_{1-42}$), total tau (tTau) and phosphorylated tau (pTau))[14] provide limited diagnostic accuracy for discriminating DLB from AD since they are also abnormal in almost 25–40% of DLB patients due to the presence of comorbid AD pathology[15–18]. Additional markers reflecting different, specific, and unique aspects of DLB pathophysiology are needed, which could be useful for different contexts of use in both clinical settings (e.g., prognosis, differential diagnosis, disease monitoring) and trials (e.g., patient selection, stratification, treatment efficacy).

CSF proteome profiling allows to identify changes covering a wide range of biological processes in vivo. As observed within the AD field,

such analysis can help to define the molecular mechanisms involved in disease pathogenesis and reveal promising biomarker candidates[19–21]. The few DLB proteomic studies performed to date did not yield many biomarker candidates, which could be due to the limited sample size (30–40 samples per group) relative to DLB heterogeneity[22–25]. We here employed a high-throughput proteomics method (immune-based proximity extension assay (PEA)) that allows analysis of large cohorts, with the additional advantage that custom multiplex immunoassays including the markers of interest can be smoothly developed for large-scale validation[21,26]. We have applied this workflow to (i) define CSF proteomic changes underlying DLB pathogenesis and (ii) to identify, develop and validate multiplex biomarker assays that could aid in the specific diagnosis of DLB.

## Results

An overview of the study design is presented in Fig. 1. We included a total of 534 participants in the discovery cohort, a subset of patients analyzed in our previous CSF proteomic study[21]. Custom multiplex panels were developed and validated in two independent clinical cohorts (validation cohort 1: $n = 164$; validation cohort 2: $n = 165$) and one AD/DLB autopsy-confirmed cohort ($n = 76$). The demographic characteristics and AD CSF biomarkers are described in Table 1. Cognitively unimpaired controls were younger in all the cohorts analyzed. Cases included in validation cohort 2 and the AD/DLB autopsy cohort were overall older than the other cohorts. The % of DLB cases with different clinical core futures was similar across clinical cohorts except for REM Sleep Behavior Disorder (RBD), which was lower in the clinical validation cohorts 1 and 2 compared to the discovery cohort. In these three cohorts, total neuropsychiatry inventory (NPI)[27,28] score was higher in DLB compared to controls. The total NPI score was also higher in DLB compared to AD in the discovery and clinical validation cohort 1, but not in the clinical validation cohort 2. The Unified Parkinson's Disease Rating Scale (UPDRS-III)[29] was similar across the clinical validation cohorts.

### CSF proteins differentially regulated in DLB

CSF proteome profiling revealed a total of 14 proteins differentially regulated in DLB compared to controls after correcting for multiple

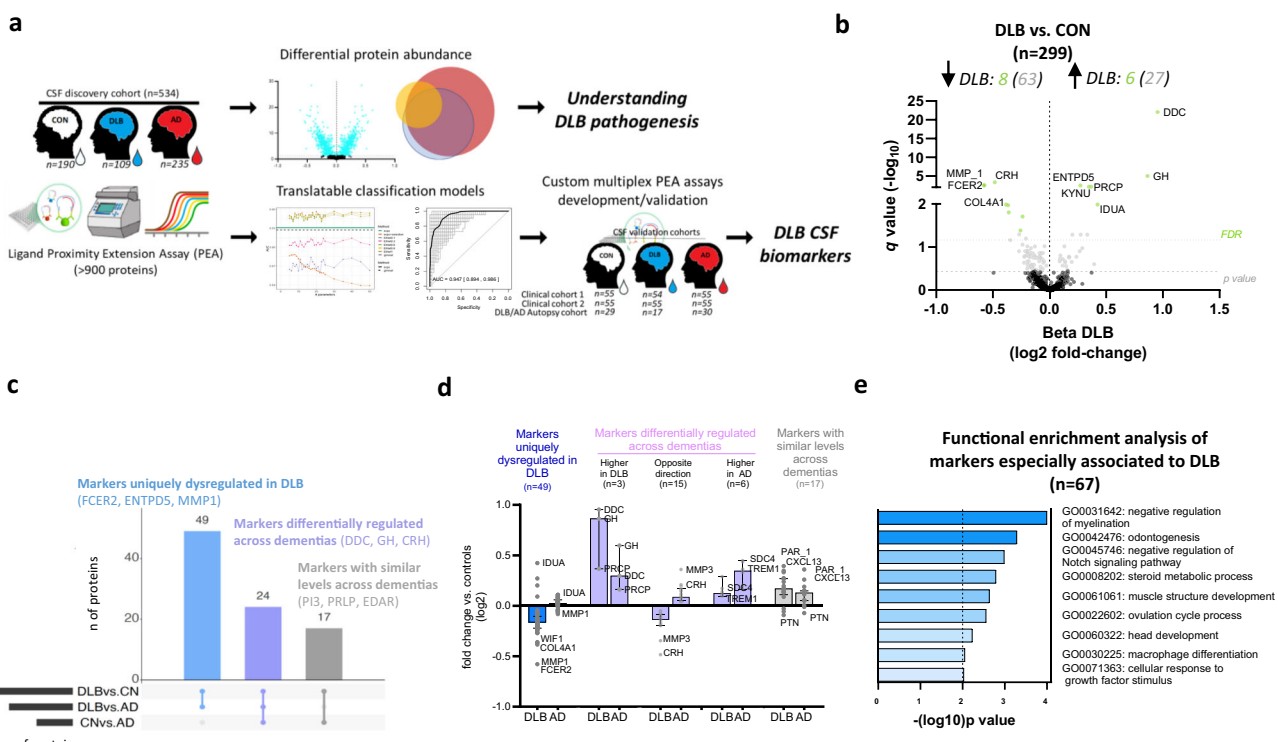

**Fig. 1 | Study overview and differential abundance of CSF proteins in DLB. a** Protein levels in CSF from cognitively unimpaired controls (white), DLB (blue) and AD (red) were measured by antibody-based PEA technology. Differential CSF protein abundance as well as classification models were investigated. Custom multiplex PEA assays containing the markers identified within the classification panels were developed and validated in three independent validation cohorts. **b** Volcano plot shows the CSF proteins that are differentially regulated in DLB vs. controls. Each dot represents a protein. The beta coefficients (log2 fold-change) are plotted versus $q$ values (−log10-transformed). Proteins significantly dysregulated after adjusting for false discovery rate (FDR, $q < 0.05$) are colored in light green and those with nominal significance ($p < 0.05$) are colored in grey. The name of the top 10 significant dysregulated CSF proteins and the top 5 with the strongest effect sizes are annotated. The total number of proteins that are down-regulated (left) or up-regulated (right) is indicated. Horizontal dotted line indicates the significance threshold. Adjusted $p$ values ($q < 0.05$) were calculated using two-sided nested linear models[21,82] adjusting for FDR[83]. **c** UpSet plot indicates which of the proteins dysregulated between DLB and controls are also dysregulated between DLB and AD or AD and controls. **d** Bar plots depict the direction of changes of the different

proteins identified when compared to controls within the subsets defined through the UpSet plot. Data are presented as median and 95% confidence interval of the fold changes of all the proteins within each subset in DLB and AD vs. CON. **e** Bar graphs depicting the biological pathways enriched in those protein dysregulated in DLB. Functional enrichment was performed using Metascape selecting Gene Ontology (GO) Biological Processes as ontology source. Terms with a $P$ value < 0.01, a minimum count of 3, and an enrichment factor >1.5 were collected and grouped into clusters based on their membership similarities. $P$ values were calculated based on the accumulative hypergeometric distribution. Kappa scores are used as the similarity metric when performing hierarchical clustering on the enriched terms, and subtrees with a similarity of >0.3 are considered a cluster. The most statistically significant term within a cluster is chosen to represent the cluster. The corresponding GO number and biological process are defined on the right side. Stronger colors represent higher significant enrichment. Vertical line represents the significant threshold (unadjusted $P$ < 0.01). CON cognitively unimpaired controls, DLB Dementia with Lewy bodies, AD Alzheimer's disease. Some images within (**a**) are courtesy of Olink® Proteomics AB.

## Table 1 | Demographic characteristics

| | Discovery cohort | | | Validation cohort 1 | | | Validation cohort 2 | | | DLB/AD autopsy cohort | | |
|---|---|---|---|---|---|---|---|---|---|---|---|---|
| | CON (n=190) | DLB (n=109) | AD (n=235) | CON (n=55) | DLB (n=54) | AD (n=55) | CON (n=55) | DLB (n=55) | AD (n=55) | CON[a] (n=29) | DLB (n=17) | AD (n=30) |
| Age, years (Mean, SD) | 58 (8)[b,c] | 69 (8)[c,d] | 66 (8)[b,d] | 58 (4)[b,c] | 69 (5)[c,d] | 66 (8)[b,d] | 62 (7)[c,d] | 76 (5)[c,d] | 72 (6)[b,d] | 63 (5)[b,c] | 76 (7)[d] | 71 (11)[d] |
| Sex (M, %) | 120 (63%) | 91 (83%) | 139 (59%) | 33 (60%) | 45 (83%) | 32 (58%) | 23 (41%) | 33 (60%) | 21 (38%) | 16 (55%) | 14 (82%) | 15 (50%) |
| MMSE (Mean, SD)[e] | 28 (2)[b] | 22 (6)[c,d] | 21 (5)[b,d] | 28 (1)[b,c] | 23 (4)[c,d] | 20 (3)[b,d] | 29 (1)[b,c] | 22 (5)[d] | 22 (4)[d] | 30 (1)[c] | 17(7) | 14 (8)[d] |
| APOE4 (+/n, %)[f] | 47/186 (25%) | 52/97 (54%) | 134/226 (59%) | 15/55 (27%) | 28/51 (55%) | 39/55 (71%) | 14/55 (25%) | 18/54 (33%) | 24/54 (44%) | na | na | na |
| CSF Aβ42, pg/mL[g] | 1121 (218)[b,c] | 764 (372)[c,d] | 603 (123)[b,d] | 1283 (301)/ 1629 (372)[b] | 851 (528)/ 751 (338)[c,d] | 566 (154)/ 494 (159)[b,d] | 1194 (503)[b,c] | 672 (454)[c,d] | 537 (206)[b,d] | 1108 (284)[b,c] | 461 (398)[d] | 404 (206)[d] |
| CSF tTau, pg/mL[g] | 211 (95)[c] | 297 (193)[d] | 746 (431)[b,d] | 292 (244)/182 (91)[d] | 293 (165)/ 214 (108)[d] | 460 (352)/ 341 (115)[b,d] | 257 (125)[b,c] | 432 (486)[c,d] | 740 (439)[b,d] | 203 (91)[b,c] | 368 (314)[d] | 535 (420)[d] |
| CSF pTau, pg/mL[g] | 38 (15)[b,c] | 52 (28)[c,d] | 92 (36)[b,d] | 54 (28)/15 (8)[c] | 48 (23)/16 (12)[c,d] | 63 (44)/36(22)[b,d] | 38 (21)[b,c] | 63 (80)[c,d] | 117 (84)[b,d] | 47 (19)[c] | 57 (28) | 72 (38)[d] |
| **Core clinical features DLB** | | | | | | | | | | | | |
| Fluctuation (+/n;%) | na | 33/43 (77%) | na | na | 28/34 (82%) | na | na | 27/37 (73%) | na | na | na | na |
| Visual Hallucinations (n/total) | na | 33/50 (66%) | na | na | 23/41 (56%) | na | na | 24/38 (63%) | na | na | na | na |
| RBD (n/total) | na | 31/37 (84%) | na | na | 20/38 (53%) | na | na | 21/36 (58%) | na | na | na | na |
| Parkinsonism (n/total) | na | 40/50 (80%) | na | na | 32/41 (78%) | na | na | 39/40 (98%) | na | na | na | na |
| NPI_total[h] | 6 (13)[b] | 12 (19)[c,d] | 9 (12)[b] | 7.5 (14)[b] | 15 (22)[c,d] | 10 (12)[b] | 0 (1)[b,c] | 20 (23)[d] | 16 (28)[d] | na | na | na |
| UPDRS-III (total)[i] | na | na | na | na | 23 (11) | na | na | 21 (12) | 2 (3) | na | na | na |

Data are median (interquartile range) unless otherwise specified. Between-group analyses were performed using two-sided univariate analysis of variance or Pearson's chi-squared test in normally distributed data with Bonferroni post-hoc adjustment. Analysis of covariance was performed for CSF biomarker analysis adjusting for age and sex when appropriate. Non-Gaussian distributed data were analyzed using Kruskal–Wallis test. Adjustment for multiple testing was performed using the Bonferroni method.

*DLB* Dementia with Lewy Bodies, *AD* Alzheimer's disease, *SD* standard deviation, *M* Male.

[a]CON group within the AD/DLB autopsy cohort are not autopsy confirmed.

[b]p < 0.05 vs.DLB.

[c]p < 0.05 vs. AD.

[d]p < 0.05 vs.CON.

[e]MMSE score was used as a measure of cognitive function it was missing for 12, 2, 6, and 41 participants in the discovery, validation 1, 2, and autopsy cohorts respectively. Between-group analyses were performed using normally distributed data with Bonferroni post-hoc adjustment. Note that there was MMSE from only 2 control cases in the autopsy cohort.

[f]APOE status was missing for 25, 3 and 2 participants in the discovery, validation 1 and 2 cohorts, respectively.

[g]Biomarker data coming from Luminex analysis in the discovery cohort was transformed using appropriate Passing–Babock transformation formulas (see methods). In validation cohort 1 the first values correspond to data obtained with Innotest and the second values correspond to data obtained with Elecsy. CSF Biomarker data from validation cohort 2 was obtained with Lumipulse.

[h]NPI_total was used as a total measure of supportive neuropsychiatry symptoms as and was available for 118 Controls 54 DLB cases and 181 AD cases in the discovery cohort; 46 controls, 50 DLB cases and 45 AD cases in validation cohort 1; 45 controls, 40 DLB cases and 38 AD cases in validation cohort 2.

[i]Total UPDRS-III was used as a measure of parkinsonism and was available for 19 DLB cases in validation cohort 1; 12 DLB cases and 4 AD cases in validation cohort 2.

testing (Fig. 1b, Supplementary Dataset 1, $q < 0,05$). Six of these proteins were upregulated in DLB (DDC, GH, IDUA, PRCP, KYNU, and ENTPD5) and eight proteins were downregulated (CRH, FCER2, MMP1, COL4A1, WIF1, PAM, VEGFA and CTSC, Fig. 1b, Supplementary Dataset 1). Of note, up to 90 proteins showed nominal significant differences between DLB and controls ($p < 0.05$; 27 upregulated and 63 downregulated in DLB Fig. 1b, Supplementary Dataset 1). Three of these proteins had replicates measured across different panels within the proteomic platform (see methods), which highly correlated with each other ($r > 0.6$; Supplementary Fig. 1). The top 5 differentially regulated proteins (median $q$: $4.37^{-04}$) are involved in the biosynthesis of dopamine and serotonin (DDC, or so-called AADC), growth control (GH), corticotropin release from pituitary gland (CRH), immune function (FCER2) and extracellular remodeling (MMP1). DDC showed the strongest effects ($\beta = 0.95$; fold-change: 1.9; $q = 8.08^{-23}$) followed by GH, MMP1, FCER2, and CRH (fold-changes >1.5; Fig. 1b and Supplementary Dataset 1).

To understand whether the protein changes identified were specific to DLB, we next analyzed if the proteins showing nominal differences in DLB were also differentially changed between DLB and AD as well as between AD and controls (Supplementary Dataset 1). UpSet plot indicates that up to 49 proteins (55%) were uniquely dysregulated in DLB (e.g., FCER2, ENTPD5, MMP1, Fig. 1c), which were mostly downregulated (Fig. 1d). Up to 17 proteins (19%) did not differ between dementias (e.g., PI3, PRLP, EDAR), which likely represent general dementia markers. A total of 24 proteins (27%) were changed between DLB and AD, but also between AD and controls (e.g., DDC, GH, CRH). When compared to controls, we observed that most of these markers showed opposite direction of changes in DLB and AD (e.g., CRH, MMP3), three markers had higher levels in DLB (e.g., DDC, GH, PRCP) and six markers had higher levels in AD (e.g., SCD4, TREM1, Fig. 1d). Functional enrichment analysis showed that the markers especially associated to DLB pathophysiology (i.e., those that were specifically changed in DLB as well as those that showed opposite direction or higher differences compared to the AD group, $n = 67$) were reflecting different biological processes including myelination regulation, tooth development, Notch signaling, steroid metabolism, muscle structure development or ovulation cycle (Fig. 1e).

## CSF proteins discriminate DLB from controls and AD dementias

DDC, the strongest dysregulated marker that is increased in DLB, could discriminate DLB from controls with high accuracy (AUC 0.91, 95% CI: 0.88–0.94, Fig. 2a). DDC could also discriminate DLB from AD with good but lower performance (AUC 0.81, 95% CI: 0.76–0.86; Fig. 2a). To investigate whether a minimal combination of biomarkers could discriminate DLB from AD patients with higher accuracies, we next performed classification analysis, followed by internal cross-validation (CSF panels, Fig. 1a). We identified a panel of 7 CSF proteins including DDC that discriminated DLB from controls and AD with higher accuracies than DDC alone (DLB vs CN AUC: 0.95, 95% CI: 0.89–0.99, DeLong´s test $p < 0.001$; DLB vs AD: AUC: 0.93, 95% CI: 0.86–0.98, DeLong´s test $p < 0.0001$, Fig. 2b, c). The model contained proteins that were dysregulated in DLB compared to both controls and AD (DDC, FCER2, CRH), as mentioned above, as well as one with nominal significant differences (MMP-3; Fig. 2c). The model also included proteins that were not changed in DLB but were specifically upregulated in AD (ABL1, MMP-10 and THOP1; Fig. 2c), as previously reported in our PEA-AD CSF study[21]. It is worth noting that similar accuracies were obtained when DLB patients with a positive or negative AD CSF biomarker profile (based on tau/Aβ42 ratio) were analyzed separately, indicating that AD pathology comorbidities did not influence the performance of the model (Supplementary Fig. 2). Sensitivity analysis using only cases from which parkinsonian medication information

was known (190 Controls (no medication), 73 DLB cases (only 13 had parkinsonian medication)) showed that CSF FCER2 was decreased only in those DLB cases undergoing treatment (Supplementary Fig. 3a). Despite medication moderately influenced the abundance of CSF DDC and CRH, these markers were still dysregulated in DLB patients without any parkinsonian treatment (Supplementary Fig. 3a). Noteworthy, both DDC and the protein panel discriminate DLB cases with no medication from controls and AD with similar performance to that obtained with the complete cohort (Supplementary Fig. 3b, c). When compared to the CSF biomarkers used to support AD diagnosis (i.e., Aβ42/tTau), we observed that the performance of the CSF panel could better discriminate DLB from controls and showed similar AUCs for the discrimination of DLB from AD (Fig. 2d). We performed correlation analysis in the complete discovery cohort to understand how these markers relate to cognitive function (MMSE) or classical AD biomarkers (Fig. 2e). Unfortunately, data on the levels of CSF α-syn in these samples was not available. For the subset of markers especially dysregulated in DLB, we observed that DDC and MMP-3 negatively correlated with MMSE, albeit weakly. Moderate positive correlations were observed between CSF (p)Tau levels and CRH and MMP-3. Weak negative correlations were detected between Aβ42 and DDC levels. As expected, the strongest correlations with AD CSF biomarkers and MMSE were observed for those markers that were specifically dysregulated in AD (ABL1, MMP10, and THOP1).

The proteins involved in the 7 CSF biomarker panel are related to different pathways including dopamine biosynthesis (DDC), immune function (FCER-2), intra and extracellular remodeling (MMP-3 and MMP-10, ABL1), regulation of the hypothalamic–pituitary–adrenal axis (CRH) and neuropeptide degradation (THOP1).

## Development and validation of custom multiplex PEA CSF protein panels

To validate the performance of our discovery findings, we developed custom multiplex PEA panels measuring six out of the seven proteins from the DLB diagnostic panel, including DDC. Custom assays showed low coefficients of variation (mean intra- and inter-assay CVs of 5% and 9% respectively) and >90% detectability (Supplementary Table 1). We next analyzed three independent CSF cohorts using these custom assays. We observed that the protein fold changes between DLB and controls or AD dementia obtained in the three validation cohorts correlated highly and in the same direction with those obtained in the discovery cohort ($r > 0.70$, Fig. 3a). Of note, the effect size of DDC change in the clinical validation cohort 2 was lower to the ones obtained with the discovery and clinical validation cohort 1 and the AD/DLB autopsy cohort (Fig. 3a). In the three cohorts, the DLB CSF panel showed slightly higher accuracies than DDC alone in discriminating DLB from controls and AD (Fig. 3b). The performance of both DDC and the panel in the clinical validation cohort 1 were similar to those observed in the discovery phase (AUCs > 0.86; Fig. 3b). In the second validation cohort, the accuracies were lower compared to those obtained in the discovery and the other validation cohorts, especially when discriminating DLB from AD (AUC_{DDC} DLBvs.CON: 0.81; AUC_{DDC} DLBvs.AD: 0.59; AUC_{custom panel} DLBvs.CN: 0.90; AUC_{custom panel} DLBvs.AD: 0.68; Fig. 3b). Sensitivity analysis in this second validation cohort showed that the classification accuracies did not improve when only DLB cases with abnormal DAT scan were analyzed (Supplementary Fig. 4a). However, analysis including only DLB cases with RBD showed similar performances to those detected in the discovery cohort (Supplementary Fig. 4b). Analysis of the AD/DLB autopsy cohort with these multiplex custom assays reported similar high accuracies as those of the discovery and clinical validation cohort 1 (AUC_{DDC} aDLBvs.CON: 0.95; AUC_{DDC} DLBvs.AD: 0.86; AUC_{custom panel} aDLBvs.CN: 1.00; AUC_{custom panel} aDLBvs.aAD: 0.90; Fig. 3b).

 

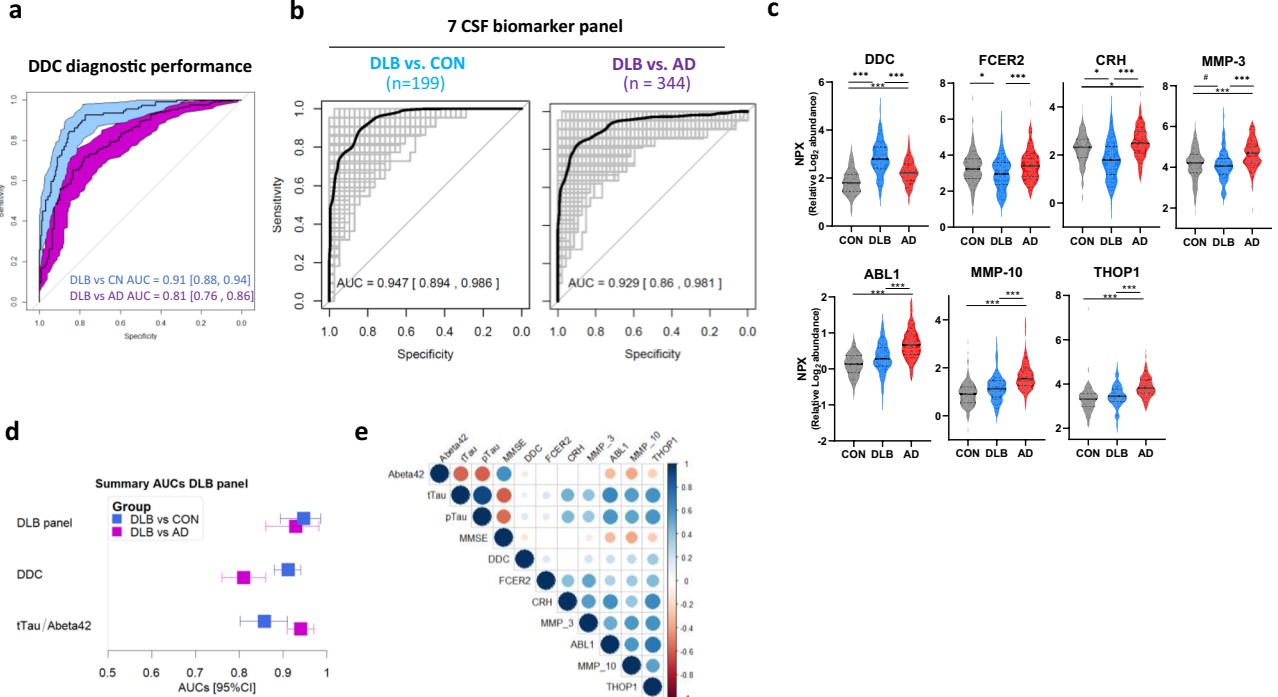

**Fig. 2 | CSF biomarker panel for specific diagnosis of DLB. a** Receiver operating characteristic (ROC) curves depict the performance of CSF DDC discriminating DLB ($n = 109$) from controls ($n = 190$, blue) and AD ($n = 235$, purple). Inset indicate the total area under the curve (AUC) and shaded areas depict 95% CI after 100 bootstrap. **b** ROC curves depicting the performance of 7 CSF biomarker panel discriminating DLB from controls and AD. Black line is the mean AUC over all re-samplings (1000 repeats of 5-fold cross-validation, gray lines). Inserts outline corresponding AUC and 95% CI. **c** Violins represent the abundance (log2 NPX) of the different CSF within the DLB biomarker panel. Horizontal black and dash lines indicate median and interquartile range of the protein abundance. **d**) Forest plot depicts the mean AUC over all re-samplings and 95% CI obtained with the CSF DLB biomarker panel, CSF DDC or CSF tTau/Abeta42 biomarkers in the comparison between DLB ($n = 109$) and controls ($n = 190$, blue) or AD ($n = 235$, purple). **e** Correlation matrix heatmap representing the Spearman's correlation coefficient in-between the proteins selected of the CSF DLB panel, the classical AD CSF biomarkers and ratios and MMSE score. Significant associations are depicted by circles. *$q < 0.05$, **$q < 0.01$, ***$q < 0.001$. DLB dementia with Lewy Bodies, AD Alzheimer's disease, CON cognitively unimpaired controls.

## CSF DDC associates with UPDRS-III, α-syn brain pathology and DLB stages

We next performed an exploratory analysis to investigate potential associations of CSF DDC and DLB-related markers (FCER2, CRH, MMP3) with different measures of DLB pathophysiology when available (i.e., UPDRS-III, post-mortem brain α-syn load, α-syn Braak stage[30] and DLB stage[31]). We observed that CSF DDC positively correlated with parkinsonism severity as measured by UPDRS-III in the clinical validation cohort 1 ($r = 0.76$; $p < 0.001$), but not in the clinical validation cohort 2 ($r = 0.33$, $p = 21$; Fig. 4a), which might be explained by the moderate changes of CSF DDC in this second validation cohort. In relation to the neuropathological data available, we observed that CSF DDC positively correlated with α-syn load in very specific brain areas (amygdala, substantia nigra, and medulla oblongata) in the autopsy confirmed cases from the discovery cohort and the AD/DLB autopsy validation cohort. In the latter, we observed additional associations with other brain areas, and thus CSF DDC also correlated with the overall total and neocortical α-syn load (Fig. 4b). In line with these results, we observed that CSF DDC increased across DLB stages in both cohorts (Fig. 4c); and along α-syn Braak stages (data available only for AD/DLB autopsy validation cohort; Fig. 4d). The levels of CSF FCER2, CRH and MMP3 also associated to different DLB pathophysiological features but these were not consistent across different cohorts (Fig. 4b and Supplementary Fig 5).

## CSF proteins within the DLB panel are dysregulated in PD

Considering that some of the DLB markers correlate with α-syn load, we next analyzed whether these markers were also dysregulated in PD using the publicly available CSF PEA-proteomic data from the Parkinson's Progression Markers Initiative[32] (Supplementary Table 2) In the AMP-PD data set, the CSF markers within the panel related to DLB (i.e., DDC, FCER2, CRH and MMP3) were all significantly dysregulated or showed nominal significance in both the prodromal and symptomatic phase of PD compared to controls (Fig. 5a). Both DDC and DLB CSF panel could discriminate proPD or PD from controls with optimal performance (AUC_DDC proPD vs. CON: 0.80; AUC_DDC PD vs. CN: 0.83; AUC_panel proPD vs. CN: 0.87; AUC_panel PD vs. CN: 0.90; Fig. 5b). In the second PPMI data set, only CSF DDC was increased in PD compared to controls and the overall performance of CSF DDC and panel was slightly reduced (AUC_DDC PDvs.CN: 0.71, AUC_panel PD vs. CN: 0.84; Fig. 5b). Please note that prodromal PD do not undergo dopaminergic medication, further supporting that the changes observed between groups are not driven by parkinsonian related treatment.

## Discussion

The DLB CSF proteome profiling performed in this study identifies different proteins specifically changed in DLB in the context of neurodegenerative dementias. We translated these findings into a six CSF protein custom panel that discriminated DLB patients from cognitively unimpaired controls and AD with high accuracies (AUCs ≥ 0.90), which was validated in independent external cohorts, including one with neuropathology confirmation. The DLB-related markers were also dysregulated in the prodromal and symptomatic phase of patients with PD, further supporting that the proteins identified reflect known biological processes associated with DLB pathophysiology such as the biosynthesis of dopamine or α-synuclein proteopathy.

Biofluid-based biomarkers specifically associated with DLB are strongly needed not only to improve timely diagnosis and diagnostic

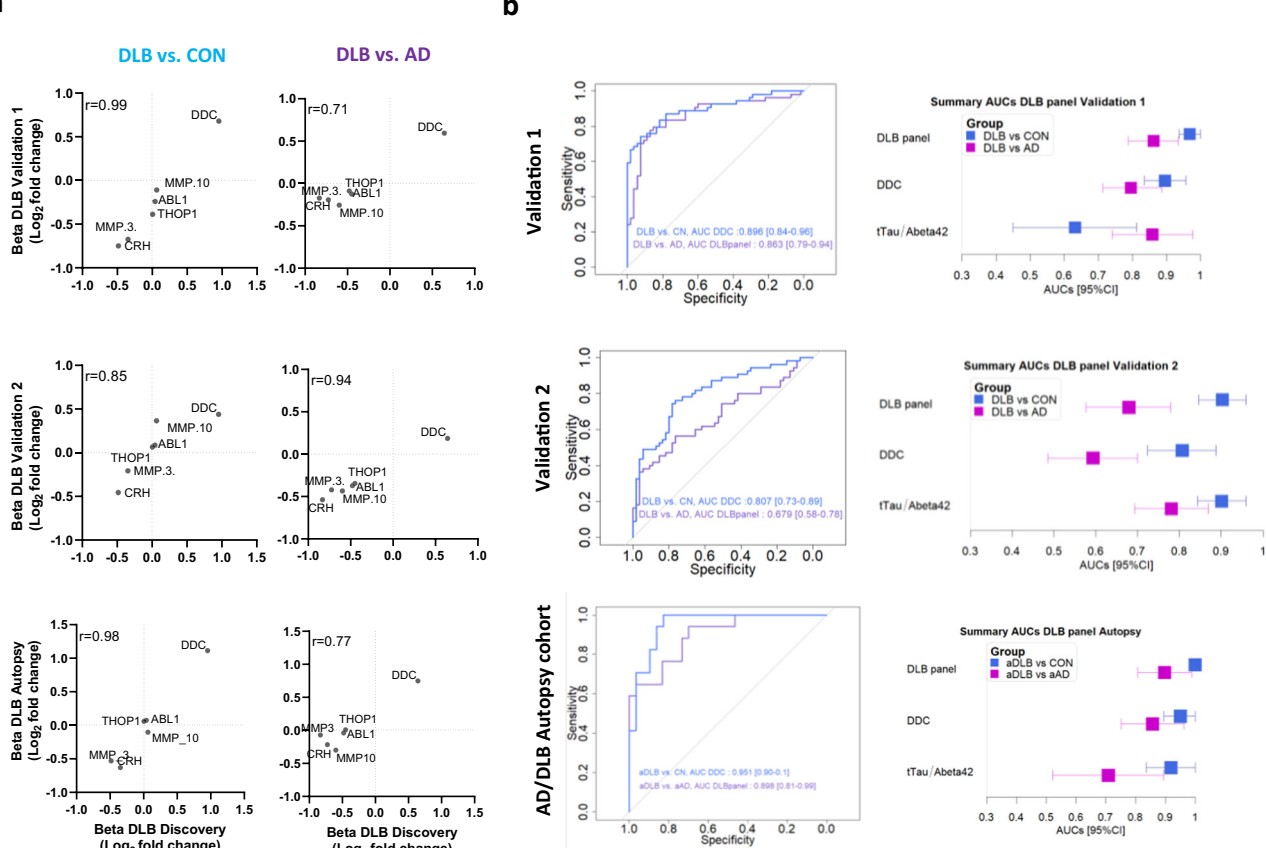

**Fig. 3 | Development and validation of custom CSF biomarker panels for DLB diagnosis in independent cohorts. a** Scatter plots show the correlation between the beta-coefficients obtained in the discovery phase to those obtained with the custom assays in the clinical validation cohorts 1 and 2 and the autopsy confirmed cohort. Insert indicate the spearman correlation coefficient. **b** Receiver operating characteristic (ROC) curves depicting the performance of DDC or the CSF biomarker panel discriminating DLB from controls or AD respectively using the custom assays across the different validation cohorts (Validation cohort 1: 54 DLB, 55 AD, and 55 CON, clinical validation cohort 2: 55 DLB, 55 AD, and CON and the AD/ DLB autopsy cohort: 17 aDLB, 30 aAD and 29 non-autopsy confirmed controls). Inserts outline the total AUC and 95% CI after 100 bootstrap. Forest plots depict the total AUC and 95%CI after 100 bootstrap obtained with the CSF DLB biomarker panel, CSF DDC, or the CSF tTau/Abeta42 ratio in the comparison between DLB and controls (blue) or AD (purple). DLB dementia with Lewy Bodies, AD Alzheimer's disease, aDLB autopsy confirmed DLB, aAD autopsy confirmed AD, CON cognitively unimpaired controls.

accuracy, but also to monitor the different biological mechanisms involved in DLB pathophysiology and as surrogate markers for clinical trials[33]. To this end, in this large CSF proteome study, we analyzed more than 100 CSF samples from DLB and cognitively unimpaired controls, but also samples from patients with AD dementia. We detected up to 90 proteins dysregulated in the CSF of DLB cases, but only 14 survived correction for multiple testing. Larger sample sizes might still be required to detect additional biomarker candidates due to clinical heterogeneity in DLB. Thanks to the inclusion of the AD dementia group, we could show that up to 55% of these 90 proteins were dysregulated in DLB specifically (e.g., FCER2, MMP1, WIF1). We identified an additional subset of CSF proteins (27%) that were differentially regulated in both DLB and AD patients compared to controls, but with different protein abundance also between DLB and AD. While some of these were decreased in DLB and increased in AD (e.g., CRH and MMP3), some were especially dysregulated in AD as previously reported (e.g., SDC4, TREM1)[21], and some were more prominently dysregulated in DLB (e.g., DDC, GH). These last shared but different protein profiles might be explained by the clinicopathological overlap across these two dementias[13,34]. Most of the proteins identified were downregulated in DLB compared to controls, a pattern observed in a previous proteome study[35] but also in recent DLB transcriptomic studies[36–39]. It has been suggested that such a strong downregulated pattern in DLB could be due to demyelination processes[38]. In line with

those findings, we observed that the proteins dysregulated in DLB were especially enriched in processes associated with negative regulation of myelination. Furthermore, previous research indicates that α-syn can induce myelin loss in neurons and oligodendroglia (precursor) cells[40,41]. Importantly, α-syn-induced myelination deficits are involved in the development of multiple system atrophy[41]. It would be thus of interest to investigate how the identified proteins relate to potential magnetic resonance imaging abnormalities and the importance of myelination processes in the pathophysiology of DLB. Of note, the three top proteins dysregulated in DLB (i.e., DDC, CRH, and GH) can regulate the hypothalamic-pituitary-adrenal (HPA) axis[42,43], suggesting that one of the major systems associated with stress response and behavioral dysfunction might be involved in DLB pathophysiology. This is in line with previous studies showing hypothalamic dysfunction in DLB[44,45].

The strongest dysregulated CSF protein in the DLB group was DDC, an enzyme involved in the biosynthesis of dopamine, which lends biological support to our biomarker discovery design[46]. This marker was not identified in previous transcriptomics studies. However, these analyses were performed in the anterior cingulate cortex[37,38], an area with limited DDC expression[47]. It should also be noticed that genetic expression is not per se a proxy reflecting protein abundance[48], which might be affected by other post-translational factors (e.g., protein clearance, protein interaction). The use of different technologies (e.g.,

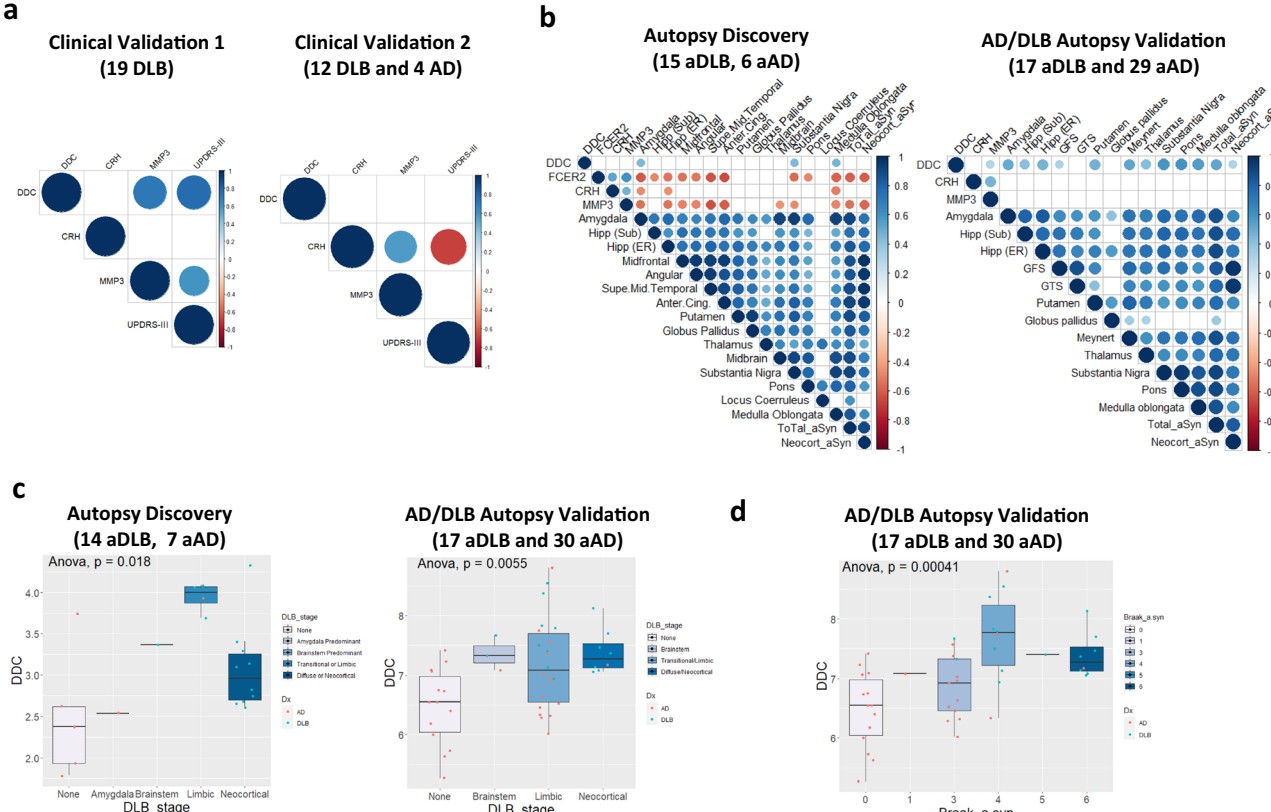

**Fig. 4 | CSF proteins within the DLB biomarker panel associate with different pathophysiological features of DLB. a, b** Correlation matrix heatmap representing the Pearson's correlation in-between the DLB proteins within the panel and (**a**) UPDRS-III in the clinical validations 1 and 2; or (**b**) α-syn load in different brain areas from a subset of cases with pathological confirmation in the discovery cohort and the AD/DLB autopsy validation cohort. Total α-syn represent the average of α-syn load across different areas. Neocortical α-syn depicts the average α-syn load in all the cortical areas (discovery: Midfrontal, angular and superior mid-temporal and anterior cingulate; Validation cohort: GFS and GTM). Significant associations are depicted by circles. **c, d** Box plots represent the abundance (log2 NPX) of CSF DDC across (**c**) the different DLB stages in the subset of cases with pathological confirmation in the discovery cohort and in the AD/DLB autopsy validation cohort; and (**d**) α-syn Braak stages in the AD/DLB autopsy cohort. Insert indicate *p*-value of one-side ANOVA analysis. Data are presented as median, upper and lower quartiles defining the interquartile range (IQR), and whiskers defining the highest and lower values without outliers (within 1.5 times IQR). Each dot represents an individual sample. DLB dementia with Lewy Bodies, AD Alzheimer's disease, aDLB autopsy confirmed DLB, aAD autopsy confirmed AD, Hipp (Sub)/(ER) Hippocampus CA/subiculum and Enthorinal, Anter. Cing. Anterios cingulate, GFS superior frontal girus, GTS superior temporal girus.

unbiased MS approaches or targeted protein arrays not containing DDC) together with the limited sample size may also explain why DDC was not detected in the few CSF proteomics studies performed so far[22–24]. Of note, we have also developed a DDC immunoassay that have shown similar changes in the clinical validation cohort 1, supporting the robustness of the findings (Bolsewig et al., in preparation). Previous studies have shown that serum DDC enzyme activity is elevated in patients using levodopa with peripheral decarboxylase inhibitors (PDI), underpinning the effect of dopaminergic treatment on DDC activity[49]. It is important to note that the increase of CSF DDC levels detected in this study are likely not driven by PDI treatment as i) PDIs do not cross the blood-brain barrier and can thus not influence DDC activity/levels in the brain and ii) we obtained similar results when only DLB patients that did not have any parkinsonism medication were included in the analysis (*n* = 60). Dysfunction of the dopaminergic system due to nigrostriatal degeneration is a well-established pathophysiological feature of DLB[50]. Previous studies have shown that nigrostriatal degeneration as well as antagonist of dopamine receptors increase DDC mRNA and activity in different models[46]. These data do not only align with our findings but also suggest that the increased DDC levels might be a response to the nigrostriatal degeneration and could thus be a very relevant biomarker for DLB diagnosis and disease monitoring. We could validate the high discriminatory accuracy for DLB vs controls (AUC: 0.91) in three independent cohorts, including a

neuropathological one (AUCs 0.81–0.95). Considering that current DLB diagnostic guidelines include supportive imaging biomarkers as proxy of nigrostriatal degeneration[4], it would be of interest to specifically analyze whether CSF DDC measurements could be an alternative or complementary diagnostic test to classical imaging scans. DDC measurements have the additional advantage of being quantitative over the binary classification with α-syn seed amplification assays, meaning that, DDC measurements might be relevant to track disease staging and for monitoring treatment responses.

To translate the CSF proteome findings into practical biomarker tools for routine diagnostics and clinical trials, we applied classification analysis and identified a panel of seven CSF proteins to discriminate DLB from controls and AD dementia with high accuracy (AUC of 0.95 and 0.93 respectively). This panel combined proteins associated to DLB (e.g., DDC, CRH, MMP3, FCER2) but also proteins specifically related to AD (ABL1, MMP-10, THOP1)[51–56], likely explaining the higher performance to discriminate these two dementias compared to DDC alone. To validate these findings in independent cohorts, we successfully developed custom multiplex assays for six out of the seven selected markers. The protein effect sizes obtained with these custom assays in the three validation cohorts correlated well with those obtained in the discovery cohort (r coefficients ranging between 0.70 and 0.99), and the high discriminative values were mostly validated (AUCs ≥ 0.80), supporting the relevance and robustness of our

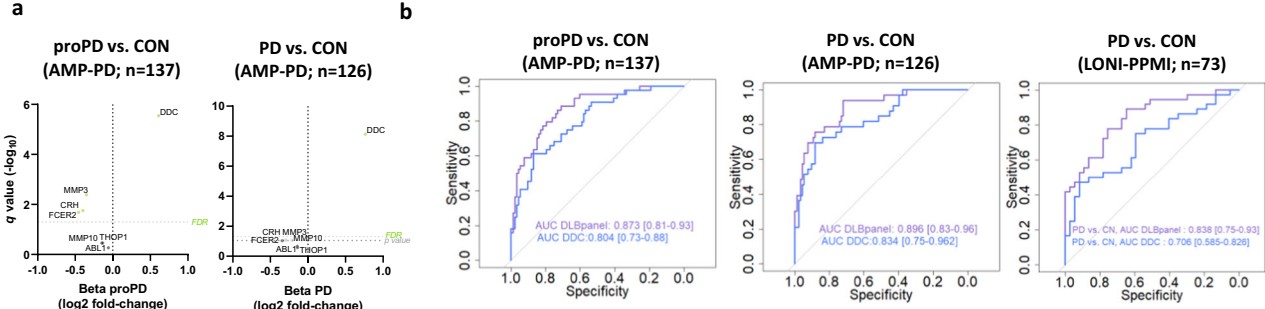

**Fig. 5 | CSF DDC and proteins within the DLB protein panel are dysregulated in the prodromal or symptomatic stages of Parkinson´s disease. a** Volcano plot shows the CSF proteins that are differentially regulated in prodromal PD or PD vs. controls. Each dot represents a protein. The beta coefficients (log2 fold-change) are plotted versus *q* values (−log10-transformed). Proteins significantly dysregulated after adjusting for false discovery rate (FDR, *q* < 0.05) are colored in light green and those with nominal significance (*p* < 0.05) are colored in gray. Horizontal dotted line indicates the significance thresholds. Adjusted *p* values (*q* < 0.05) were calculated using two-sided nested linear models[21, 82] adjusting for FDR[83]. **b** Receiver operating characteristic (ROC) curves depicting the performance of DDC (blue) or the CSF biomarker panel (purple) discriminating prodromal PD or PD from controls in the two cohorts (AMP-PD: 44proPD, 33 PD and 99 controls; and LONI-PPMI: 36 PD and 37 controls). CSF PEA proteome data is publicly available from Parkinson's Progression Markers Initiative (see method section). Inserts outline corresponding AUC and 95% CI after 100 bootstrap. PD Parkinson´s disease, pro-PD prodromal PD, CON cognitively unimpaired controls.

findings. In the clinical validation cohort 2 we observed, however, lower accuracy when discriminating DLB vs. AD (AUC 0.68), which was not dependent on whether cases were DAT positive or negative. Sensitivity analysis using only RBD-positive DLB cases in the clinical validation cohort 2 showed a similar performance to that observed in the discovery cohort. We also observed that the total NPI score, in contrast to the different scores observed for the discovery or the clinical validation cohort 1, was similar between DLB and AD cases in clinical validation cohort 2. Overall, these data suggest that the heterogeneity of the clinical diagnosis of DLB based on different supportive features (e.g., DLB cases that do not have specific core clinical features, such as RBD; or differences in neuropsychiatry symptoms, measured with NPI) may impact biomarker data; and that the panel is likely most sensitive and specific in typical DLB. Understanding which factors may influence CSF DDC levels is of paramount importance for the potential future implementation of this marker and the corresponding DLB diagnostic panel.

We could not investigate the association between CSF DDC or the other DLB markers (CSF CRH, FCER2, and MMP3) and CSF α-syn levels, as this information was not available in our data sets. Still, the added value of such information might be limited considering the variable results on CSF α-syn observed across studies[6–8]. We however explored the association between the CSF DLB markers identified in this study and specific DLB pathophysiological features. Even though these specific results should be interpreted with caution considering the limited sample size, the positive association of CSF DDC with UPDRS-III, α-syn load in different brain areas, α-syn Braak stage and DLB stages supports the potential of this marker to track disease progression already from very early stages. This is further supported by the data obtained from the PD cohorts in which CSF DDC was already increased in the prodromal stage of the disease. The other biomarkers within the DLB panel (i.e., CRH, FCER2, and MMP3) also associated with some DLB pathophysiological features, but results were not consistent across cohorts. This calls for a systematic analysis of all these CSF markers in cohorts with large associated neuropathological characterization. These markers were also dysregulated in prodromal PD and showed nominal significance at the clinical stage. The fact that the DLB markers identified in this study are also dysregulated in PD is not surprising considering both disorders are characterized by α-syn pathology and dopaminergic cell loss in the substantia nigra[57]. Thus, despite the fact that the identified markers are specific for DLB in the context of neurodegenerative dementias, they might be dysregulated in other α-synucleopathies or disorders characterized by dopamine deficiency. Overall, all these data support that CSF DDC (and maybe the panel)

could be a useful quantitative tool not only to track nigrostriatal degeneration and disease stage, but also to select prodromal cases or monitor treatment effects in clinical trials. Specific follow-up studies including different Parkinsonian disorders and Lewy bodies dementias in the same data set are needed to understand the role of these markers across the different disorders and define their potential context of use.

Despite the large number of samples and proteins analyzed in this DLB-specific study, there are still relevant limitations. We observed that Parkinsonian medication moderately influenced the levels of some CSF proteins. Despite the number of cases that underwent Parkinsonian medication was limited (*n* = 13), such data highlights the importance of including treatment information in biomarker studies. Additional analyses are needed to confirm the relevance of treatment effect on CSF FCER2 abundance. Importantly, medication did not drive the main differences observed between groups for the other DLB-related proteins (i.e., CSF, CRH). Considering the clinicopathological overlap with AD[13,34], we cannot exclude that the potential misdiagnosis of DLB patients may have influenced our discovery results. However, samples were collected in well-characterized biobanks from specialized memory units, and DLB diagnosis was either autopsy confirmed or supported by DAT scans in more than one third of the patients. Moreover, our sensitivity analysis showed similar diagnostic accuracies in DLB cases with positive AD CSF biomarker profile, and we further validated the biomarker panel in a CSF AD/DLB autopsy confirmed cohort. The similar differences observed for CSF DDC, FCER2, CRH, and MMP3 also in the PD data sets call for additional studies that evaluate the performance of these markers in other α-synucleinopathies (e.g., multiple system atrophy), other dementia types with motor dysfunctions (e.g., progressive supranuclear palsy, corticobasal degeneration) or other conditions with dopamine deficiency (e.g., psychiatry disorders). This DLB study covers already three phases of the biomarker development workflow[58] including the retrospective analysis of four cohorts coming from different memory clinics with their own intrinsic methodologies and clinical characteristics, which ultimately supports the reproducibility and robustness of the findings. Still, future studies are needed to define the clinicopathological correlations and trajectories between the biomarker panel and different measures associated with DLB pathophysiology, including other relevant markers, such as α-syn seed amplification assays[9–12,59], in longitudinal and prospective studies using independent large and highly phenotyped cohorts. This will help to define their potential context of use within different settings in clinical practice and trials (prognosis, diagnosis, monitoring, etc.). We envision that

CSF DDC and the panel developed within this study could be relevant complementary and quantitative diagnostic tests reflecting different biological aspects associated with DLB pathophysiology in the context of dementias. This quality may make them suitable to improve diagnosis and staging along the DLB continuum but also to monitor treatment response in clinical trials targeting different mechanisms[60,61].

Overall, the CSF proteome profile performed in this study identified CSF biomarkers specifically associated with DLB, providing an holistic view into the pathophysiology of this dementia. The protein panels discriminate DLB from controls and AD dementia with high accuracy, which we have translated into custom assays and validated in independent cohorts, including one with AD/DLB autopsy confirmation. These biomarkers and panels are ready to be employed to define their added value and potential context of use in clinical settings and trials within the context of DLB. The use of an antibody-based technology allowed us to overcome the cross-technology gap often encountered in biomarker studies[62] and efficiently translate our discovery findings into customized assays. Current studies are ongoing to validate the CSF biomarkers using alternative immunoanalytical platforms. The workflow employed in this study may ultimately facilitate bench-to-bedside translation of biofluid-based biomarker findings and could thus be also relevant for other research fields.

## Methods

### Ethics statement

The studies were approved by the Institutional Ethical Review Boards of each center (Discovery cohort: VUmc: AD CSF biobank METC number 00-211; University of Pennsylvania: language and cognitive impairment in Parkinson's disease and Parkinson's disease with dementia or dementia with Lewy bodies IRB069801; SPIN cohort: COLLECTION 16/2013;). Informed consent was obtained from all subjects or their authorized representatives.

### Participants

An overview of the study design is presented in Fig. 1. The total discovery cohort ($n = 534$) included CSF samples from patients diagnosed with DLB ($n = 109$), AD ($n = 235$) and 190 cognitively unimpaired controls (CON; Table 1). Most of the samples were selected from the Amsterdam Dementia Cohort (ADC) and DEvELOP[63,64]. To enrich for DLB dementia cases, additional CSF samples from the Center for Neurodegenerative Disease Research at the University of Pennsylvania were included (49 DLB and 18 AD)[65]. Three additional independent CSF cohorts were used for validation of the customized panels (see methods below): clinical validation cohort 1 (from the ADC: 54 DLB, 55 AD, and 55 controls)[63], clinical validation cohort 2 (from the Sant Pau Initiative on Neurodegeneration (SPIN) cohort: 55 DLB, 55 AD, and 55 controls)[66] and an AD/DLB autopsy confirmed cohort (from BIODEM and the neurobiobank of the Institute Born-Bunge (IIB)/UAntwerp: 17 autopsied confirmed DLB (aDLB) and 30 autopsied confirmed AD (aAD))[67,68]. An additional 29 cognitively unimpaired controls from BIODEM-UAntwerp were included in this cohort but these were not autopsy confirmed[67,68]. CSF was collected by lumbar puncture and processed and stored at all sites in agreement with the JPND-BIOMARKAPD guidelines, thereby minimizing the influence of potential pre-analytical factors[69].

All participants of every cohort underwent standard neurological and cognitive assessments and diagnosis was assigned according to international consensus criteria for DLB[31,70] and AD[71]. The neuropathological validation cohort included cases with a definite diagnosis according to international neuropathological examination guidelines for DLB[31,70] and AD[72]. The control group included individuals with subjective cognitive decline, in whom objective cognitive and laboratory investigations were normal (i.e., criteria for MCI,

dementia, or any other neurological or psychiatric disorder not fulfilled) with additionally negative AD CSF biomarkers in all cases[63,73–75]. Mini-Mental State Examination (MMSE) was used as a measure of global cognition. Motor parkinsonism was assessed using section III of the Unified Parkinson's Disease Rating Scale (UPDRS)[29]. Core and supportive clinical features[31] were recorded locally in each memory unit[63,64,66]. Evaluation of REM Sleep Behavior Disorder (RBD) includes an interview with a sleep specialist, a full nocturnal video-polysomnography, the Mayo Sleep questionnaire (cutoff ≥1) or caregivers reporting that a patient would 'act out' their dreams and moves extensively during their sleep as previously described[64,66,76]. Supportive neuropsychiatry symptoms (e.g., depression, delusions, apathy, anxiety) were compiled according to the Neuropsychiatry Inventory and summarized in one single score (NPI-total)[27,28]. Levels of CSF $A\beta_{42}$, tTau and pTau(181) ('AD CSF biomarkers') were used to support AD diagnoses. These markers were analyzed locally as part of the diagnostic procedure using commercially available kits (VUmc and UAntwerp: ELISA INNOTEST Aβ(1-42), hTAUAg, phospho-Tau(181P), Fujirebio, Ghent, Belgium) or VUmc: Aβ(1-42), t-TAUAg, phospho-Tau181 Elecsys biomarker assays (Roche Diagnostics GmbH); Penn: Luminex xMAP INNO-BIA AlzBio3; Luminex Corp, Austin, TX; SPIN: Lumipulse G600, Fujirebio)[67,77,78]. Positive CSF AD biomarker profile was defined locally as increased tTau/$A\beta_{42}$ ratio in the cohorts from ADC (>0.46) and Pennsylvania (>0.30); and low $A\beta_{42/40}$ ratio (<0.062) and high total tau (>456 pg/ml) or p-tau (>63 pg/ml) in the SPIN cohort[66,77,79,80].

In the discovery cohort, DLB neuropathology was confirmed in 14 DLB patients (13%) by autopsy (Supplementary Table 3). Clinical diagnosis was supported by FPCIT single-photon emission computed tomography (DAT scan) in 23 patients (24% of the total DLB patients with clinical diagnosis, Supplementary Table 3). A total of 48 DLB cases (44%, Supplementary Table 3) did not have autopsy data or any additional supporting biomarker information (e.g., electroencephalogram, EEG)[31]. Comorbid AD pathology, which is common in DLB patients[15–18], was present in up to 34 DLB patients (32%) as measured by AD CSF biomarker profile. From all the DLB patients, only 13 DLB patients had medication for parkinsonian symptoms, 60 did not have such a treatment and this information was not available for 36 cases.

In the clinical validation cohorts 1 and 2, DAT scans supported DLB diagnoses in 18 and 21 patients respectively (33% and 38% of the total DLB patients, Supplementary Table 3). A total of 18 and 27 DLB patients from validation cohorts 1 and 2 respectively (33% and 49%, Supplementary Table 3) did not have autopsy data or any additional supporting biomarker information (e.g. electroencephalogram, EEG)[31]. Comorbid AD pathology was present in up to 26 and 25 DLB patients from validation cohorts 1 and 2, respectively (50% and 46%, respectively).

The AD/DLB autopsy validation cohort included cases with a definite diagnosis according to international neuropathological examination guidelines for DLB[31,70] and AD[72]. Within the aDLB group coexisting AD pathological changes ($n = 7$) or cerebrovascular lesions ($n = 3$) were reported in 10 cases. Coexisting cerebrovascular lesions ($n = 8$), TDP pathology ($n = 1$) or cerebral amyloid angiopathy ($n = 1$) were reported within the aAD neuropathological group. The control group of this cohort was not autopsy confirmed. It consisted of volunteers, mainly spouses of patients who visited the memory clinic. The inclusion criteria for these volunteers were: (1) no neurological or psychiatric antecedents; (2) no organic disease involving the central nervous system following extensive clinical examination; and (3) normal neuropsychological exam. Exclusion criteria consisted of brain tumors, large cerebral infarction/bleeding, strategic infarctions, other neurodegenerative diseases, severe head trauma, epilepsy, brain infections, severe depression, unregulated diabetes mellitus, untreated thyroid disorders, or any severe somatic comorbidity that interferes with study participation[81].

Patient demographics and clinical and biochemical values from all cohorts used in this study are listed in Table 1.

## CSF protein profiling

As part of our large-scale discovery project[21], CSF proteins (979) were quantified using the 11 specific and validated multiplex antibody-based protein panels based on the proximity extension assay (PEA) that were available at the time in which the analysis was performed as previously described (Cardiometabolic, Cardiovascular II and III, cell regulation, development, immune response, inflammation, metabolism, neurology, oncology II and organ damage; Olink Proteomics, Uppsala, Sweden; Supplementary Table 4). Each panel contains reagents to measure up to 92 unique proteins, though 30 proteins can be measured in several panels (replicates). Briefly, samples were randomized across plates containing appropriate intra- and inter-plate quality controls (QC) from the manufacturer and measured in two different rounds. Each round included 16 bridging samples covering different clinical groups, which were used for reference sample normalization to control for potential batch effects. Each assay has an experimentally determined lower limit of detection (LOD) estimated as three standard deviations above the noise level from the negative controls that are included on every plate. Only proteins with values over the lower limit of detection (LOD) in at least 85% of the samples were selected for further statistical analysis, in which remaining raw values under LOD (2.4% of all measurements) were kept as provided by the manufacturer. A total of 665 proteins (642 unique proteins) were ultimately included for statistical analysis of the discovery cohort (Supplementary Table 4).

## Development of custom PEA assays

Custom-designed multiplex-PEA assays were developed by the manufacturer following standardized protocols[21,26]. We developed assays to measure six out of the seven proteins selected upon the classification analysis described in section 2.4. Besides the corresponding clinical samples, each plate included four CSF QC samples, a negative control and three calibrators used for normalization. QC samples and calibrators were measured in triplicate. Each custom assay has an experimentally determined LOD defined as for the discovery panels. Precision (intra- and inter-assay CV) were calculated using the 4 CSF QC samples. No cross-reactivity between assays for specific proteins was detected. Assay parameters including LOD, detectability and CVs are included in Supplementary Table 1. Samples from validation cohorts were randomized across plates and normalized for any plate effects using the built-in inter-plate controls according to manufacturers' recommendations. Protein abundance was reported in NPX values.

## PEA CSF proteome information from Parkinson´s disease (PD) cohort

We downloaded clinical and PEA proteome data (Olink 1536 explore; which include the following 384-panels: Cardiometabolic, Inflammation, Neurology and Oncology) from the Parkinson´s Progressive Markers Initiative PPMI[32] (http://www.ppmi-info.org/data) on March 28th, 2023. The PPMI study is registered with ClinicalTrials.gov (number NCT01141023). The study protocols and the manuals related to different procedures for clinical assessment, biochemical phenotyping as well as sample handling, collection and storage is described online (http://www.ppmi-info.org/study-design). Data from samples collected at baseline were used for the analysis. Two different sets were available with PEA CSF proteome (Supplementary Table 2): the Accelerating Medicines Partnership Parkinson´s disease (AMP-PD; 93 controls, 44 prodromal PD and 33 PD patients)) and the PPMI_set (37 controls and 36 PD). The methods and manuals regarding PEA proteome analysis as well as the quality control reports are available online.

## Statistical analysis

All data preprocessing and analyses were conducted using R version 3.5.3 and SPSS version 25. Between-group analyses for the demographic variables were performed using two-sided one-way analysis of variance in normally distributed continuous data or Pearson's chi-square test for categorical variables. Analysis of covariance was performed when an association between classical AD CSF biomarker and age and/or sex were detected. Adjustment for multiple testing was performed using Bonferroni method. Non-Gaussian distributed data were analyzed using Kruskal-Wallis Test. For the CSF proteome data, differences in protein abundance between pairs of clinical groups were evaluated using nested linear models as previously described, in which for each individual protein feature, we assessed if its addition to a base model containing age and gender contributed to model fit[21,82]. For each pairwise comparison, multiplicity was taken into account by controlling the False Discovery Rate (FDR)[83] at $q \le 0.05$ based on the number of features analyzed. We next evaluated which CSF protein combination (CSF panels) could best discriminate the groups of interest while keeping the number of markers to the minimum, so that they can be ultimately translated into small, practical custom panel[21]. For this purpose, binary classification signatures (DLB vs. CON and DLB vs. AD) were constructed by way of penalized generalized linear modeling (GLM) with an elastic net penalty (a linear combination of lasso and ridge penalties) in the discovery CSF cohort using the glmnet package and including age and sex as covariates[21,82,84]. This penalty enables estimation in settings where the feature-to-sample ratio is too high for standard generalized linear regression. Moreover, it performs automatic feature decorrelation as well as feature selection. For each classification exercise, we compare multiple models which reflect (a) a grid of values for the elastic-net mixing parameter, reflecting strong decorrelation to a pure logistic lasso regression and (b) a grid of values reflecting the maximum number of proteins that may be selected under each model (21 markers maximum). The former grid (a) considers that we have little information on the collinearity burden in the data. The latter grid (b) considers that we want to keep the number of selected proteins relatively low for the future development of customized panels. The optimal penalty parameters in the penalized models were determined based on (balanced) 10-fold cross-validation of the model likelihood[21,82]. The cross-validation was performed with balanced folds, by which each fold has an outcome group ratio close to the corresponding ratio in the full data set, also referred to as stratified cross-validation. The predictive performance of all models was assessed by way of (the comparison of) Receiver Operating Characteristic (ROC) curves and Area Under the ROC Curves (AUCs). The model with the highest AUC and lowest number of markers for each classification signature was selected. The fold-based selection proportions for each marker were assessed to identify and select the most promising markers within each model (i.e., features that are stably selected across each individual fold thereby minimizing potential overfitting). To reflect the manual selection pressure for these final marker sets, each final logistic signature was subjected to a ridge-regularization with a penalty parameter of 0.1. The performance (AUC) was evaluated by internal validation: repeated 5-fold cross-validation with 1000 repeats. The 95% confidence interval around the resulting AUCs was based on resampling quantiles (percentile method). External validations assessed the performance of the final models with the markers of interest in the validation cohorts using ROC analysis.

Non-parametric correlation analysis was performed to understand the associations between the proteins within the CSF panels and the classical AD CSF biomarkers or cognitive function (MMSE score) using the complete discovery cohort without stratifying per diagnostic category and conditioning on age and sex as covariates. Innotest values generated for the Amsterdam Dementia Cohort were adjusted for drift over time as previously described[85]. UPENN values had lower means for $A\beta_{42}$ on the Innotest, which were first linearly transformed

to normalize to the same mean. Passing-Bablock transformation formulas were calculated based on individuals with both Luminex and Innotest values for Aβ$_{42}$ ($n = 32$), tTau ($n = 32$) and pTau ($n = 27$) and used the formulas to estimate the equivalent Innotest values for those samples measured with Luminex platform only (transformed_Aβ$_{42}$ = (Luminex_ Aβ$_{42}$*4.65) − 36.23; transformed_tTau = (Luminex_tTau*5.28) − 2.03; transformed_ptau = (Luminex_pTau*1.88) + 27.36). The association between the markers of interest with DLB pathophysiological features (i.e., UPDRS, α-syn load, α-syn Braak stage[30] and DLB stage[31]) was analyzed by either correlation analysis or ANOVA. Stratification or post-hoc analysis with DLB pathophysiological features could not be performed due to the limited sample size of some subgroups.

Functional enrichment analysis was performed using Metascape[86] selecting GO Biological Processes as ontology source. All the CSF proteins optimally analyzed with Olink arrays ($n = 645$ protein gene products) were included as the enrichment background. Default parameters were used for the analysis in which terms with a $p$-value $< 0.01$, a minimum count of 3, and an enrichment factor $> 1.5$ were collected and grouped into clusters based on their membership similarities.

### Reporting summary
Further information on research design is available in the Nature Portfolio Reporting Summary linked to this article.

## Data availability
The source data generated in this study are available within this study (Supplementary Dataset 1) or have been deposited in the Synapse database under the accession code https://www.synapse.org/PRIDE_DLB. The PD data used in the preparation of this article were obtained from the Parkinson's Progression Markers Initiative (PPMI) database (www.ppmi-info.org/access-data-specimens/download-data). For up-to-date information on the study, visit www.ppmi-info.org.

## Code availability
The codes and scripts used in this study have been deposited in the synapse database under accession code https://www.synapse.org/PRIDE_DLB. All models were built using publicly available packages and functions in R.

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

## Acknowledgements

The authors acknowledge Prof. Dr. Jean-Jacques Martin (Institute Born-Bunge, Antwerp) for the neuropathological characterization of the Antwerp cohort. Data used in the preparation of this article were obtained on March 28th, 2023 from the Parkinson's Progression Markers Initiative (PPMI) database (www.ppmi-info.org/access-data-specimens/download-data), RRID:SCR_006431. For up-to-date information on the study, visit www.ppmi-info.org. This research is part of the neurodegeneration research program of Amsterdam Neuroscience. Alzheimer Center Amsterdam is supported by Stichting Alzheimer Nederland and Stichting VUmc fonds. The chair of Wiesje van der Flier is supported by the Pasman stichting. The clinical database structure was developed with funding from Stichting Diorapthe. Funding from ZonMW(# 733050509),

Alzheimer Nederland and Stichting Diorapthe supported the DEvELOP study. This study was supported by Alzheimer Nederland (CT, MC), ZonMW (#73305095007), Health~Holland, Topsector Life Sciences & Health (PPP-allowance; #LSHM20106) and Instituto de Salud Carlos III (PI20/01330 and AC19/00103 to AL, PI18/00435 and INT19/00016 to DA), Fondo Europeo de Desarrollo Regional (FEDER), Unión Europea, "Una manera de hacer Europa", and CIBERNED (Program 1, Alzheimer Disease). MC is supported by the attraction talent fellowship of Comunidad de Madrid (2018-T2/BMD-11885) and "PROYECTOS I + D + I – 2020"—Retos de investigación from the Ministerio Español de Ciencia e innovación (PID2020-115613RA-I00). W.F., A.L., and C.T. are recipients of TAP-Dementia, a ZonMW funded project (#10510032120003) under the Dutch National Dementia Strategy. The collection of patient samples and data from Penn University was supported by different funding sources: National Institute on Aging (P01-AG066597), National Institute on Aging P30-AG072979 (formerly P30-AG10124), National Institute on Aging U19-AG062418-03 (formerly NINDSP50-NS053488-09)). PPMI—a public-private partnership—is funded by the Michael J. Fox Foundation for Parkinson's Research and funding partners, including 4D Pharma, Abbvie, AcureX, Allergan, Amathus Therapeutics, Aligning Science Across Parkinson's, AskBio, Avid Radiopharmaceuticals, BIAL, Biogen, Biohaven, BioLegend, BlueRock Therapeutics, Bristol-Myers Squibb, Calico Labs, Celgene, Cerevel Therapeutics, Coave Therapeutics, DaCapo Brainscience, Denali, Edmond J. Safra Foundation, Eli Lilly, Gain Therapeutics, GE HealthCare, Genentech, GSK, Golub Capital, Handl Therapeutics, Insitro, Janssen Neuroscience, Lundbeck, Merck, Meso Scale Discovery, Mission Therapeutics, Neurocrine Biosciences, Pfizer, Piramal, Prevail Therapeutics, Roche, Sanofi, Servier, Sun Pharma Advanced Research Company, Takeda, Teva, UCB, Vanqua Bio, Verily, Voyager Therapeutics, the Weston Family Foundation and Yumanity Therapeutics.

## Author contributions

M.C. and C.T. conceived and designed the study. M.C., L.V., C.P., and M.v.N. performed the statistical analysis. M.C., L.V., Y.H., A.S., A.L., D.A., S.E., J.V.A., S.A., A.C.-P., D.J.I., W.M.F., A.W.L., and C.T. recruited participants and collected clinical data and samples. M.C. and Y.H. arranged and prepared samples for proteomics analysis. M.C. and C.T. drafted de manuscript. All authors contributed to the revision and editing of the manuscript.

## Competing interests

M.C. has been an invited speaker at Eisai, is an associate editor at Alzheimer´s Research & Therapy and has been an invited writer for Springer Healthcare. L.V. received a grant for CORAL consortium by Olink. D.I. is a Scientific Advisory Board Member for Denali Therapeutics. D.A. participated in advisory boards from Fujirebio-Europe and Roche Diagnostics and received speaker honoraria from Fujirebio-Europe, Roche Diagnostics, Nutricia, Krka Farmacéutica S.L., Zambon S.A.U. and Esteve Pharmaceuticals S.A. D.A., and A.L. declare a filed patent application (Title: Markers of synaptopathy in neurodegenerative disease; Applicant: Fundació Institut de Recerca de l'Hospital de la Santa Creu i Sant Pau, Inventors: Olivia BELBIN; Alberto LLEÓ; Alejandro BAYÉS; Juan FORTEA; Daniel ALCOLEA; Application number: PCT/EP2019/056535; International Publication Number: WO 2019/175379 A1; Current status: Active. Licensed to ADx Neurosciences NV (Ghent, Belgium); This patent is not related to any specific aspect of the current manuscript). W.F. has performed contract research for Biogen MA Inc, and Boehringer Ingelheim. W.F. has been an invited speaker at Boehringer Ingelheim, Biogen MA Inc, Danone, Eisai, WebMD Neurology (Medscape), Springer Healthcare. W.F. is consultant to Oxford Health Policy Forum CIC, Roche, and Biogen MA Inc. WF participated in advisory boards of Biogen MA Inc and Roche. All funding is paid to her institution. W.F. is a member of the steering committee of PAVE, and Think Brain Health. WF was associate editor of Alzheimer, Research & Therapy in 2020/2021. W.F. is an associate editor at Brain. C.E.T. has a collaboration contract with ADx Neurosciences,

Quanterix and Eli Lilly, performed contract research or received grants from AC-Immune, Axon Neurosciences, Bioconnect, Bioorchestra, Brainstorm Therapeutics, Celgene, EIP Pharma, Eisai, Grifols, Novo Nordisk, PeopleBio, Roche, Toyama, Vivoryon. She serves on editorial boards of Medidact Neurologie/Springer, Alzheimer Research and Therapy, Neurology: Neuroimmunology & Neuroinflammation, and is editor of a Neuromethods book Springer. She had speaker contracts for Roche, Grifols, Novo Nordisk. The rest of the authors declare no competing interest.

## Additional information

[1]Neurochemistry Laboratory and Biobank, Department of Clinical Chemistry, Amsterdam Neuroscience, Amsterdam University Medical Centers, Location VUmc, Amsterdam, The Netherlands. [2]Barcelonaβeta Brain Research Center, Pasqual Maragall Foundation, Barcelona, Spain. [3]Departamento de Ciencias Farmacéuticas y de la Salud, Facultad de Farmacia, Universidad San Pablo-CEU, CEU Universities, Madrid, Spain. [4]Alzheimer Center Amsterdam, Department of Neurology, Amsterdam Neuroscience, Amsterdam University Medical Centers, Location VUmc, Amsterdam, The Netherlands. [5]Mathematical & Statistical Methods group (Biometris), Wageningen University & Research, Wageningen, The Netherlands. [6]Lab of neuropathology, Neurobiobank, Institute Born-Bunge, Antwerp University, Edegem, Belgium. [7]Department of Neurology, Institut d'Investigacions Biomèdiques Sant Pau (IIB SANT PAU) - Hospital de Sant Pau, Universitat Autònoma de Barcelona, Hospital de la Santa Creu i Sant Pau, Barcelona, Catalunya, Spain. [8]Center of Biomedical Investigation Network for Neurodegenerative Diseases (CIBERNED), Madrid, Spain. [9]Department of Epidemiology & Data Science, Amsterdam Public Health research institute, Amsterdam University Medical Centers, Location VUmc, Amsterdam, The Netherlands. [10]Reference Center for Biological Markers of Dementia (BIODEM), Laboratory of Neurochemistry and Behavior, Institute Born-Bunge, University of Antwerp, Antwerp, Belgium. [11]Vrije Universiteit Brussel, Center for Neurosciences (C4N), Neuroprotection and Neuromodulation Research Group (NEUR), Brussels, Belgium. [12]Universitair Ziekenhuis Brussel, Department of Neurology, Brussels, Belgium. [13]Department of Neurology, Perelman School of Medicine, University of Pennsylvania, Philadelphia, PA, USA. ✉e-mail: mcampo@barcelonabeta.org

