## [Peer Review File · Nature Communications]

REVIEWER COMMENTS

Reviewer #1 (Remarks to the Author):

Comments for the manuscript “CSF proteome profiling reveals highly specific biomarkers for dementia with Lewy bodies”

Using proximity extension assay (PEA) as previously reported (del Campo, M. et al., Nat Aging, 2022), authors identified multiple CSF proteins dysregulated in DLB, which were especially related to myelination processes and an enzyme involved in dopamine biosynthesis (L-amino acid decarboxylase, DDC) was the most dysregulated protein in DLB and could discriminate DLB from controls and AD patients. Based on the discovery analysis, authors developed 7 CSF proteins panel containing DDC as a major component, which exhibited the slight improvement of the discrimination power compared to the assay with DDC only. Authors tried to validate the performance of finally-selected 6 CSF proteins panel in three independent cohorts. Here, I saw the following two critical points are missing in the current manuscript to conclude author’s statement that the panel can be employed to diagnose DLB and define the context-of-use in clinical settings (prognosis, diagnosis, and monitoring).

1. In the clinical validation cohort 2, authors observed much lower accuracy when discriminating DLB vs. AD (AUC 0.68) which may not be effective to diagnose DLB in symptomatic population. Although authors mentioned that the heterogeneity of the clinical diagnosis may explain the differences observed across the three validation cohorts, there are no discussion to explain the detail of heterogeneity in each validation cohort in the current manuscript. Unless authors justify the discrepancy of the clinical validation cohort 2 in the manuscript clearly, it would be hard to say the proposed CSF proteins-panel was fully validated to diagnose DLB because these three validation cohorts were set and analyzed as prospective way with pre-defined objective for the biomarker validation.

2. In this manuscript, most of readouts to discuss the biomarker performance are described as a binary manner such as ROC-AUC. This enables to discuss the diagnostic accuracy (+/-) of DLB, however does not allow discussing the potential use for disease staging and monitoring treatment responses (- described in manuscript line 397-399 and 425-426). Authors showed the correlation between each CSF biomarker and MMSE which may be relevant to the disease severity of DLB (Fig 2e) even though it is not the perfect clinical measure for DLB. In this figure, no or quite-low correlations can be seen between DCC (and other CSF proteins identified as DLB-relevant) and MMSE, suggesting the CSF biomarker panel proposed in this manuscript has less potential to track the disease stage and/or monitoring? It is necessary to describe the most practical way to use the proposed biomarker for DLB based on the data presented exactly. If the potential utilities for DLB staging and monitoring are discussed in the manuscript, the additional analyses (I suggest in the following comments as examples) would be necessary.

Also, there are several major comments/suggestions for improving the manuscript as followings.

1. In the Discovery cohort, clinical validation cohort 1, and clinical validation cohort 2, “DAT-SPECT scan” data are present for some cases although the N is not so high. If the quantitative data (not just a visual read) are available for DAT-SPECT scan, it is interesting to see the quantitative correlation between the imaging and the CSF DCC (and the proteins panel) to discuss if the proposed biomarker can be used for staging and monitoring purposes.
2. In the Discovery cohort and Autopsy cohort, some samples are “autopsy confirmed CSF”. Here, if the detail of neuropathology information are available such as Braak staging for alpha-synuclein pathology, authors should map the neuropathology information together with the CSF proteins results to consider if the proposed biomarker can be used for staging and monitoring purposes. It could be important to discuss the staging potential for the proposed CSF protein panel in the quantitative manner, not only ROC-AUC – which could become the differentiation point from the existing biomarker assay such as RT-QuIC.
3. If authors explain the biology of the selected 6 CSF proteins as linking to “myelination” due to alpha-synuclein pathology, it is highly recommended to discuss the correlation with the diffusion MRI imaging (e.g., DTI or DWI) if the data are available. That may support if the DCC-containing protein panel works to stage DLB as well as diagnose or not.

Followings are minor comments.

1. The most important and novel finding in this study would be “CSF DCC” as DLB biomarker. This was elucidated by the non-biased proteomics approach using PEA platform for CSF samples from DLB, AD, and control participants. Meanwhile, there are some other studies to explore the changes in DLB brain and biofluids using other omics approaches. As authors referenced, e.g., gene expression profiling of brain samples from DLB patients was extensively conducted (Pietrzak, M. et al., *Biochem Biophys Res Commun*, 2016) and it has been hypothesized that e.g., demyelination processes are highly involved in the DLB pathophysiology. In these previous studies, it seems that the change of DCC gene expression was not reported in human DLB brains, which makes bit-puzzling. It would be better to further describe the hypothetical link of “CSF DCC change” and “brain DCC change” to explain the background biology to support the proposed biomarker for DLB diagnosis in addition to the current description.
2. Was the soluble alpha-synuclein (SNCA, H6UYS0) monitored in the discovery panel of PEA assay and what was the readout? – although I guess there was no change as previously reported. It is good to touch this point briefly because alpha-synuclein is the proximal protein to DLB pathology although the general soluble species may not be relevant to the pathology – like CSF total tau.

Reviewer #2 (Remarks to the Author):

The authors sought to use proximity extension-based multiplex assays to establish the specific CSF proteomic changes that underlie DLB in a well-characterized cohort of DLB patients, AD patients and controls. They applied this workflow to define novel CSF proteomic changes underlying DLB pathogenesis and to identify, develop and validate multiplex biomarker assays that could aid in the specific diagnosis of DLB.

This paper has many strengths, and the data are both timely and impressive. The methodology is generally sound (but see comments below). The findings have face validity. The stated limitations are appropriate, and the suggested next steps for analyses are appropriate.

This paper would be strengthened if a few issues could be addressed:

Readability and consistency – the paper is challenging to read. The cohorts are different, and slightly different methods for determining diagnoses were used. For example, validation cohort 1 had 35% of AD patients who were CSF pTau negative. Also, “four out of 13 of the DLB patients with autopsy confirmation had also a positive AD CSF profile” – does this mean that path-proven Lewy body disease had a positive AD CSF profile despite the absence of AD pathology? These differing cohort characteristics represent weakness as well, and should be acknowledged. The cohorts could be better described so that readers are clear about the cohort characteristics.

Terminology – like many investigators, the abbreviations DLB and AD are being used in this paper for non-autopsied patients based on clinical features and biomarkers, and the same abbreviations are being used for autopsied cases. The terms should be distinctly different to allow readers to better differentiate cohorts defined by clinical features and antemortem biomarkers, and those defined by pathology.

Applicability – the findings have diagnostic as well as pathophysiologic and trial implications, and this reviewer finds the pathophysiologic and clinical trial use implications more important than the diagnostic ones. For example, the seeding assays (RT-QuIC) for CSF for alpha-synuclein already show high sensitivity and specificity, and if CSF is going to be collected in the clinic as part of a DLB diagnostic assessment, clinicians would likely order RT-QuIC as well as a proteomics profile as is shown here. Do the authors also plan to apply RT-QuIC to at least a subset of these samples to compare the utility? As the authors state, the pathophysiologic implications of the specific proteins identified in their panel is important and may lead to more insights on pathogenesis. Also, the quantifiable nature of one or more proteins may be particularly useful as outcomes in clinical trials more so than RT-QuIC. But analyses on longitudinally collected CSF samples are needed to see what changes, if any, occur in the proteins of interest. This latter point is implied but not specifically stated – it should be stated outright. Please elaborate on these issues more in the discussion.

Reviewer #3 (Remarks to the Author):

This manuscript is about the development of a panel of protein biomarkers in spinal fluid in Lewy Body Dementia (DLB). The initial cohort consists in 109 DLB patients, 235 patients with Alzheimer's and 190 controls without dementia from two centres. Six found markers were independently validated in two further cohorts with 55 controls and Alzheimer and 54 DLB in cohort 1 and an independent cohort 2 with 55 each controls, Alzheimer and DLB and an autopsy cohort of 76 but not all of these had an autopsy.

I have a few comments:

Since LBD is a disease between Alzheimer and also Parkinson's this should have been validated also in CSF samples from PD subjects. The PD rating scale should be given in parallel to the cognitive measures as DLB is a mix of cognitive decline and PD motor symptoms.

The cohort descriptions are a bit difficult to follow about DaT Scans, AD CSF signatures etc. The main marker, that could have supported the clinical diagnosis well would have been this new synuclein RT-Quic.

In table 1 it is obvious, that the controls were youngest in all 4 cohorts.

The controls of the "autopsy cohort" were not autopsied and the number and name as autopsy cohort is therefore not correct.

Also, it would have been good to see the independent validation of the identified panel by other assays (e.g. ELISA or MS) and not by the same platform that is still an antibody assay, that may identify changes upon the antibody combination. The antibody combination should be written.

DDC seems to be a marker protein, which is likely changed due to dopaminergic medication in the LBD group. In general, the medication and the UPDRS (as mentioned above) should be given in table 1 and also the levodopa equivalent dosage. The explanation that even in a subset of 55 DLB subjects, that were not on medication, DDC was increased is limited as dopaminergic medication could also be taken if necessary and not continuously.

Minor:

Often in the manuscript the direction is not given and writing that there is a change in this and that marker does lack this information on the direction.

There should be more discussion on the pathophysiology of the other markers than DDC and possibly some pathway analysis.

Reviewer #1 (Remarks to the Author):

Comments for the manuscript “CSF proteome profiling reveals highly specific biomarkers for dementia with Lewy bodies”

Using proximity extension assay (PEA) as previously reported (del Campo, M. et al., Nat Aging, 2022), authors identified multiple CSF proteins dysregulated in DLB, which were especially related to myelination processes and an enzyme involved in dopamine biosynthesis (L-amino acid decarboxylase, DDC) was the most dysregulated protein in DLB and could discriminate DLB from controls and AD patients. Based on the discovery analysis, authors developed 7 CSF proteins panel containing DDC as a major component, which exhibited the slight improvement of the discrimination power compared to the assay with DDC only. Authors tried to validate the performance of finally-selected 6 CSF proteins panel in three independent cohorts. Here, I saw the following two critical points are missing in the current manuscript to conclude author’s statement that the panel can be employed to diagnose DLB and define the context-of-use in clinical settings (prognosis, diagnosis, and monitoring).

1. In the clinical validation cohort 2, authors observed much lower accuracy when discriminating DLB vs. AD (AUC 0.68) which may not be effective to diagnose DLB in symptomatic population. Although authors mentioned that the heterogeneity of the clinical diagnosis may explain the differences observed across the three validation cohorts, there are no discussion to explain the detail of heterogeneity in each validation cohort in the current manuscript. Unless authors justify the discrepancy of the clinical validation cohort 2 in the manuscript clearly, it would be hard to say the proposed CSF proteins-panel was fully validated to diagnose DLB because these three validation cohorts were set and analyzed as prospective way with pre-defined objective for the biomarker validation.

Based on reviewer’s comment, we have now looked further into different potential factors that may explain the heterogeneity of the cohorts, including clinical (e.g., presence of core clinical symptoms, neuropsychiatry symptoms; NPI), motor parkinsonism (e.g., section III of the Unified Parkinson Disease Rating Scale; UPDRS-III) and preanalytical (e.g., sample handling and storage). Demographic and preanalytical factors are likely not influencing the data, as they are similar to the other cohorts. We observed that the number of DLB cases that had REM Sleep Behavior Disorder (RBD) in the clinical validation cohort 2 was lower compared to the discovery cohort (53 vs 84% respectively), but similar to that of the clinical validation cohort 1. Sensitivity analysis using only RBD positive DLB cases in the clinical validation cohort 2 showed similar performance to that observed in the discovery cohort. We also observed that the total NPI score, in contrast to the different scores observed for the discovery or the clinical validation cohort 1, was similar between DLB and AD cases in clinical validation cohort 2. Overall, these data further support that the heterogeneity of the clinical phenotypes (e.g., DLB cases that do not have specific core clinical features, e.g., RBD; or differences in neuropsychiatry symptoms, measured with NPI) may affect the biomarker performance, and that the panel is likely most sensitive and specific in typical DLB. All this information has now been added as follows:

1. Methods, page 4: *“Motor parkinsonism was assessed using section III of the Unified Parkinson’s Disease Rating Scale (UPDRS)¹. Core and supportive clinical features² were recorded locally in each memory unit³⁻⁵. Supportive neuropsychiatry symptoms (e.g., depression, delusions, apathy, anxiety) were compiled according to the Neuropsychiatry Inventory and summarized in one unique score (NPI-total)^{6,7}. “*
2. Results:

- a. Page 9 : *“The % of DLB cases with different clinical core features was similar across clinical cohorts except for REM Sleep Behavior Disorder (RBD), which was lower in the clinical validation cohorts 1 and 2 compared to the discovery cohort. In these three cohorts, total NPI score was higher in DLB compared to controls. Total NPI score was also higher in DLB compared to AD in the discovery and clinical validation cohort 1, but not in the clinical validation cohort 2. UPDRS was similar across the clinical validation cohorts.”*
 - b. Page 12: *“Sensitivity analysis in this second validation cohort showed that the classification accuracies did not improve when only DLB cases with abnormal DAT scan were analyzed (Supplementary figure 4a). However, analysis including only DLB cases with RBD showed similar performances to those detected in the discovery cohort (Supplementary figure 4b)”*
3. Discussion: page 15: *“In the clinical validation cohort 2 we observed, however, lower accuracy when discriminating DLB vs. AD (AUC 0.68), which was not dependent on whether cases were DAT positive or negative. Sensitivity analysis using only RBD positive DLB cases in the clinical validation cohort 2 showed similar performance to that observed in the discovery cohort. We also observed that the total NPI score, in contrast to the different scores observed for the discovery or the clinical validation cohort 1, was similar between DLB and AD cases in clinical validation cohort 2. Overall, these data suggest that the heterogeneity of the clinical diagnosis of DLB based on different supportive features (e.g., DLB cases that do not have specific core clinical features, such as RBD; or differences in neuropsychiatry symptoms, measured with NPI) may impact biomarker data; and that the panel is likely most sensitive and specific in typical DLB.”*

This unique and comprehensive discovery and validation DLB study covers already three phases of the biomarker development workflow⁸, in which four different cohorts have been analyzed. We agree with the reviewer that additional studies in large and thoroughly phenotyped cohorts are however still needed to understand how these biomarkers can be fully validated and implemented in clinical settings. We have now added this information in the limitation section as follows: *“This DLB study covers already three phases of the biomarker development workflow⁸ including the retrospective analysis of four cohorts coming from different memory clinics with their own intrinsic methodologies and clinical characteristics, which ultimately supports the reproducibility and robustness of the findings. Still, future studies are needed to define the clinicopathological correlations and trajectories between the biomarker panel and different measures associated with DLB pathophysiology, including other relevant markers, such as RT-QuIC α -syn⁹⁻¹³, in longitudinal and prospective studies using independent large and highly phenotyped cohorts. This will help to define their potential context of use within different settings in clinical practice and trials (prognosis, diagnosis, monitoring, etc.)”*

2. In this manuscript, most of readouts to discuss the biomarker performance are described in a binary manner such as ROC-AUC. This enables to discuss the diagnostic accuracy (+/-) of DLB, however does not allow discussing the potential use for disease staging and monitoring treatment responses (- described in manuscript line 397-399 and 425-426). Authors showed the correlation between each CSF biomarker and MMSE which may be relevant to the disease severity of DLB (Fig 2e) even though it is not the perfect clinical measure for DLB. In this figure, no or quite-low correlations can be seen between DCC (and other CSF proteins identified as DLB-relevant) and MMSE, suggesting the CSF biomarker panel proposed in this manuscript has less potential to track the disease stage and/or monitoring? It is necessary to describe the most practical way to use the proposed biomarker for DLB based on the data presented exactly. If the potential utilities for DLB staging and monitoring are discussed in the manuscript, the additional analyses (I suggest in the following comments as examples) would be necessary.

We agree that MMSE might not be the most optimal clinical measure of disease stage/severity in DLB, which may explain the low correlations. We thank the reviewer for the suggestions provided below to explore the

potential of the identified biomarkers to track disease stage or monitor treatment response. For subset of cases, we managed to compile part of the suggested clinical and pathological data, including the severity of motor parkinsonism using UPDRS-III scores, as suggested also by reviewer 3. We observed that CSF DDC positively correlated with UPDRS-III in the clinical validation cohort 1, but not in the clinical validation cohort 2 ($r = 0.33$, $p = 0.21$), which might be explained by the moderate changes of CSF DDC detected in this second validation cohort (figure 4a). Despite the limited sample size of the datasets containing additional clinical and pathological information, the preliminary analyses overall support the potential use of especially CSF DDC to track disease progression, next to supporting DLB diagnosis from a biological perspective (see below). All this information is now included in a new results paragraph “3.4 CSF DDC associates to UPDRS-III, α -syn brain pathology and DLB stages.” Follow-up studies specifically designed to answer these questions in larger data sets should however be performed to confirm these results.

Also, there are several major comments/suggestions for improving the manuscript as followings. 1. In the Discovery cohort, clinical validation cohort 1, and clinical validation cohort 2, “DAT-SPECT scan” data are present for some cases although the N is not so high. If the quantitative data (not just a visual read) are available for DAT-SPECT scan, it is interesting to see the quantitative correlation between the imaging and the CSF DCC (and the proteins panel) to discuss if the proposed biomarker can be used for staging and monitoring purposes.

Semi-quantitative DAT-SPECT data (i.e., based on striatal binding ratio) was available for only 11 DLB cases within the validation cohort 1. Within this limited data set, no correlation was observed between the markers of interest and the semiquantitative DAT values in any of the specific brain areas analyzed. However, considering the high variability and the presence of outliers, this data set does not have enough power to evaluate such associations. This information has not been added into the manuscript to avoid confusion, but we could do so if the reviewer or the editor find it informative.

Reviewers figure 1. Correlation of CSF DDC of the proteins within the DLB panel with semiquantitative DAT scan data. Semiquantitative DAT scan data (based on striatal binding ratio; Darcourt et al, 2010. European Journal of Nuclear medicine and Molecular Imaging, 2010) for different brain areas was available for 11 DLB patients from the clinical validation cohort 1. Dot plot depicts the Pearson correlations between CSF DDC (y-axis) and the total average DAT semiquantitative score (x-axis). Correlation matrix heatmap depicts the pearson's correlation coefficient in-between the different CSF proteins within the DLB CSF biomarker panel and semi-quantitative DAT values for each brain area. Significant associations are depicted by circles. Stria: striatum, caud: caudate nucleus, put: putamen.

2. In the Discovery cohort and Autopsy cohort, some samples are “autopsy confirmed CSF”. Here, if the detail of neuropathology information are available such as Braak staging for alpha-synuclein pathology, authors should map the neuropathology information together with the CSF proteins results to consider if the proposed biomarker can be used for staging and monitoring purposes. It could be important to discuss the staging potential for the proposed CSF protein panel in the quantitative manner, not only ROC-AUC – which could

become the differentiation point from the existing biomarker assay such as RT-QuIC. We have gathered the data available in the different biobanks and performed additional staining to complete the neuropathological information. We collected neuropathological data from a subset of cases of the discovery cohort (15 DLB and 7 AD cases) and autopsy validation cohort (17 DLB and 30 AD cases), which included a-syn load in different brain regions, DLB stages and a-syn Braak stage (the latter only available for the autopsy validation cohort). The results can be summarized as follows:

1. CSF DDC: DDC positively correlated with a-syn load in very specific brain areas (amygdala, substantia nigra and medulla oblongata) in both cohorts. In the autopsy validation cohort, we observed additional associations of DDC with a-syn load in other brain areas, and thus it also correlated with the overall total and neocortical a-syn load (figure 4b). In line with these results, we observed that CSF DDC increased across DLB stages in both cohorts (figure 4c); and along a-syn Braak stages (data available only for autopsy validation cohort; figure 4d).
2. Other DLB CSF related proteins (CRH, FCER2, MMP3): the abundance of these proteins also associated to different DLB pathophysiological features, but these were not consistent across different cohorts (Figure 4b and supplementary fig 5), calling for additional studies in larger neuropathological data sets.

These results have been included in the new section “3.4 CSF DDC associates to UPDRS-III, a-syn brain pathology and DLB stages.” and figure 4 and supplementary figure 5.

3. If authors explain the biology of the selected 6 CSF proteins as linking to “myelination” due to alpha-synuclein pathology, it is highly recommended to discuss the correlation with the diffusion MRI imaging (e.g., DTI or DWI) if the data are available. That may support if the DCC-containing protein panel works to stage DLB as well as diagnose or not.

We would like to clarify that myelination was one of the pathways identified in the functional enrichment analysis using the 90 proteins that were dysregulated in DLB based on nominal significance. Unfortunately, we could not access to MRI imaging data. However, we think that the data gathered with previous points (UPDRS, neuropathological data) provide sufficient preliminary data supporting the potential added value of CSF DDC and DLB related proteins for disease staging. We have now highlighted the importance of follow-up studies using MRI imaging data in the discussion (page 14): *“It would be thus of interest to investigate how the identified proteins relate with potential magnetic resonance imaging abnormalities and the importance of myelination processes in the pathophysiology of DLB.”*

Followings are minor comments.

1. The most important and novel finding in this study would be “CSF DCC” as DLB biomarker. This was elucidated by the non-biased proteomics approach using PEA platform for CSF samples from DLB, AD, and control participants. Meanwhile, there are some other studies to explore the changes in DLB brain and biofluids using other omics approaches. As authors referenced, e.g., gene expression profiling of brain samples from DLB patients was extensively conducted (Pietrzak, M. et al., *Biochem Biophys Res Commun*, 2016) and it has been hypothesized that e.g., demyelination processes are highly involved in the DLB pathophysiology. In these previous studies, it seems that the change of DCC gene expression was not reported in human DLB brains, which makes bit-puzzling. It would be better to further describe the hypothetical link of “CSF DCC change” and “brain DCC change” to explain the background biology to support the proposed biomarker for DLB diagnosis in addition to the current description. In agreement with reviewer’s comment we have added the following explanations in the discussion (page 14): *“This marker was not identified in previous transcriptomics studies. However, these analyses were performed in the anterior cingulate cortex^{9,10}, an area with limited DDC expression¹¹. It should also be noticed that genetic*

expression is not per se a proxy reflecting protein abundance¹², which might be affected by other post-translational factors (e.g., protein clearance, protein interaction). The use of different technologies (e.g., unbiased MS approaches or targeted protein arrays not containing DDC) together with the limited sample size may also explain why DDC was not detected in the few CSF proteomics studies performed so far¹³⁻¹⁵. Of note, we have also developed a DDC immunoassay that have shown similar changes in the clinical validation cohort 1, supporting the robustness of the findings (Bolsewig et al, in preparation). ”

2. Was the soluble alpha-synuclein (SNCA, H6UYS0) monitored in the discovery panel of PEA assay and what was the readout? – although I guess there was no change as previously reported. It is good to touch this point briefly because alpha-synuclein is the proximal protein to DLB pathology although the general soluble species may not be relevant to the pathology – like CSF total tau. Unfortunately, CSF alpha-synuclein was not within the PEA arrays by the time we performed this study. This information was included as follows:

- Results (page 11): *“Unfortunately, data on the levels of CSF a-synuclein in these samples was not available.”*
- Discussion (page 16): *“We could not investigate the association between CSF DDC or the other DLB markers (CSF CRH, FCER2 and MMP3) and CSF α -syn levels. Still, the added value of such information might be limited considering the variable results on CSF α -syn levels observed across studies¹⁶⁻¹⁸.”*

Reviewer #2 (Remarks to the Author):

The authors sought to use proximity extension-based multiplex assays to establish the specific CSF proteomic changes that underlie DLB in a well-characterized cohort of DLB patients, AD patients and controls. They applied this workflow to define novel CSF proteomic changes underlying DLB pathogenesis and to identify, develop and validate multiplex biomarker assays that could aid in the specific diagnosis of DLB.

This paper has many strengths, and the data are both timely and impressive. The methodology is generally sound (but see comments below). The findings have face validity. The stated limitations are appropriate, and the suggested next steps for analyses are appropriate.

We thank the reviewer for the positive feedback.

This paper would be strengthened if a few issues could be addressed:

Readability and consistency – the paper is challenging to read. The cohorts are different, and slightly different methods for determining diagnoses were used. For example, validation cohort 1 had 35% of AD patients who were CSF pTau negative. Also, “four out of 13 of the DLB patients with autopsy confirmation had also a positive AD CSF profile” – does this mean that path-proven Lewy body disease had a positive AD CSF profile despite the absence of AD pathology? These differing cohort characteristics represent weakness as well, and should be acknowledged. The cohorts could be better described so that readers are clear about the cohort characteristics.

We agree with the reviewer that the explanation of the different cohorts is complex, partly due to some information irrelevant for the interpretation of the findings (e.g., exhaustive information about classical AD CSF biomarkers in the different cohorts). We have now reorganized the methods section and summarize the information relevant for the main message (e.g., number of DLB cases with AD co-pathology).

The slight differences on the methodologies used to characterize the cohorts is to some extent expected considering samples come from different centers worldwide and the heterogeneity of the disease. We acknowledge that a more systematic analysis in large and highly phenotype cohorts is needed to understand how the different DLB related markers associate to the DLB pathophysiology. Still, we also think that the use of different cohorts from different international memory units support the robustness of the results and pave the way for potential future implementation in clinical settings.

This has now been elaborated more in the limitation section: *“This DLB study covers already three phases of the biomarker development workflow⁸ including the retrospective analysis of four cohorts coming from different memory clinics with their own intrinsic methodologies and clinical characteristics, which ultimately supports the reproducibility and robustness of the findings. Still, future studies are needed to define the clinicopathological correlations and trajectories between the biomarker panel and different measures associated with DLB pathophysiology, including other relevant markers, such as RT-QuIC α -syn^{19–23}, in longitudinal and prospective studies using independent large and highly phenotyped cohorts^{820–24}.”*

Terminology – like many investigators, the abbreviations DLB and AD are being used in this paper for non-autopsied patients based on clinical features and biomarkers, and the same abbreviations are being used for autopsied cases. The terms should be distinctly different to allow readers to better differentiate cohorts defined by clinical features and antemortem biomarkers, and those defined by pathology.

In agreement with reviewers’ comments, autopsied cases have been now named autopsy confirmed DLB and AD (aDLB and aAD respectively).

Applicability – the findings have diagnostic as well as pathophysiologic and trial implications, and this reviewer finds the pathophysiologic and clinical trial use implications more important than the diagnostic ones. For example, the seeding assays (RT-QuIC) for CSF for alpha-synuclein already show high sensitivity and specificity, and if CSF is going to be collected in the clinic as part of a DLB diagnostic assessment, clinicians would likely order RT-QuIC as well as a proteomics profile as is shown here. Do the authors also plan to apply RT-QuIC to at least a subset of these samples to compare the utility? As the authors state, the pathophysiologic implications of the specific proteins identified in their panel is important and may lead to more insights on pathogenesis. Also, the quantifiable nature of one or more proteins may be particularly useful as outcomes in clinical trials more so than RT-QuIC. But analyses on longitudinally collected CSF samples are needed to see what changes, if any, occur in the proteins of interest. This latter point is implied but not specifically stated – it should be stated outright. Please elaborate on these issues more in the discussion.

We have now added preliminary data supporting the use of the markers identified in this study to monitor disease staging, and thereby possibly treatment effects, suggesting these could be additional quantitative tools complementary RT-QuIC analysis (see points 1-3 reviewer one). RT-QuIC is a relatively novel technology that have shown highly promising findings^{19,20,23}. Still, it is laborious and requires further optimization before its implementation in research and clinical settings²⁴. Thus, RT-QuIC α -syn measurements have not been performed yet in any of the cohorts included in this study. We agree that these aspects should be investigated further in prospective and longitudinal cohorts. We have addressed this in the discussion:

- *Page 16: “Even though these specific results should be interpreted with caution considering the limited sample size, the positive association of CSF DDC with UPDRS-III, α -syn load in different brain areas, α -syn Braak stage and DLB stages supports the potential of this marker to track disease progression already from very early stages. This is further supported by the data obtained from the PD cohorts in which CSF DDC was already increased in the prodromal stage of the disease.”*
- *Page 16: “Overall, all these data support that CSF DDC (and maybe the panel) could be a useful quantitative tool not only to track nigrostriatal degeneration and disease stage, but also to select prodromal cases or monitor treatment effects in clinical trials. Specific follow-up studies including different parkinsonian disorders and Lewy bodies dementias in the same data set are needed to understand the role of these markers across the different disorders and define their potential context-of-use.”*
- *Page 17: “Still, future studies are needed to define the clinicopathological correlations and trajectories between the biomarker panel and different measures associated with DLB pathophysiology, including other relevant markers, such as RT-QuIC α -syn^{19–23}, in longitudinal and prospective studies using independent large and highly phenotyped cohorts”*

Reviewer #3 (Remarks to the Author):

This manuscript is about the development of a panel of protein biomarkers in spinal fluid in Lewy Body Dementia (DLB). The initial cohort consists in 109 DLB patients, 235 patients with Alzheimer's and 190 controls without dementia from two centres. Six found markers were independently validated in two further cohorts with 55 controls and Alzheimer and 54 DLB in cohort 1 and an independent cohort 2 with 55 each controls, Alzheimer and DLB and a autopsy cohort of 76 but not all of these had a autopsy.

I have a few comments:

Since LBD is a disease between Alzheimer and also Parkinson's this should have been validated also in CSF samples from PD subjects.

We thank the reviewer for their comments. We agree that measurement of the CSF biomarkers in PD cohorts is highly interesting considering the pathological overlap of DLB with parkinsonian related disorders and the now added data supporting a relation with α -syn neuropathological stages. To this end, we now analyzed an Olink proteomic dataset from prodromal and clinical PD CSF samples previously generated by the Parkinson's Progression Markers Initiative (PPMI). In this dataset we observed that DDC and the DLB related markers of the CSF panel (i.e., MMP3, FCER2, CRH) were also altered in both the prodromal and the disease stage of PD, supporting the association of the proteins identified to alpha-synuclein pathology and dopamine dysfunction. In agreement with reviewer's request, this new data has now been included in a specific paragraph in the results section ("*3.5 DDC and DLB related proteins within the CSF panel are dysregulated in the prodromal and symptomatic stage of Parkinson's disease*"; figure 5) and discussed as follows (page 16,17):

- *"Even though these specific results should be interpreted with caution considering the limited sample size, the positive association of CSF DDC with UPDRS-III, α -syn load in different brain areas, α -syn Braak stage and DLB stages supports the potential of this marker to track disease progression already from very early stages. This is further supported by the data obtained from the PD cohorts in which CSF DDC was already increased in the prodromal stage of the disease."*
- *"The fact that the DLB markers identified in this study are also dysregulated in PD is not surprising considering both disorders are characterized by α -syn pathology and dopaminergic cell loss in the substantia nigra²⁵. Thus, despite the fact that the identified markers are specific for DLB in the context of neurodegenerative dementias, they might be dysregulated in other α -synucleinopathies or disorders characterized by dopamine deficiency. Overall, these data support that CSF DDC (and maybe the panel) could be a useful quantitative tool not only to track nigrostriatal degeneration and disease stage, but also to select prodromal cases or monitor treatment effects in clinical trials. Specific follow-up studies including different parkinsonian disorders and Lewy bodies dementias in the same data set are needed to understand the role of these markers across the different disorders and refine their potential context-of-use."*
- *"The similar differences observed for CSF DDC, FCER2, CRH and MMP3 also in the PD data sets call for additional studies that evaluate the performance of these markers in other α -synucleinopathies (e.g., multiple system atrophy), other dementia types with motor dysfunctions (e.g., progressive supranuclear palsy, corticobasal degeneration) or other conditions with dopamine deficiency (e.g., psychiatry disorders)."*

The PD rating scale should be given in parallel to the cognitive measures as DLB is a mix of cognitive decline and PD motor symptoms.

In agreement with reviewer's comment we have gathered this information when available and included in table 1. Interestingly, we observed that CSF DDC concentrations positively correlate with UPDRSIII in the clinical validation cohort 1, but not in the second validation cohort. This new data has been now included and discussed:

- Results (page 12): “We observed that CSF DDC positively correlated with parkinsonism severity as measured by UPDRS-III in the clinical validation cohort 1 ($r=0.76$; $p < 0.001$), but not in the clinical validation cohort 2 ($r=0.33$, $p=21$, figure 4a), which might be explained by the moderate changes of CSF DDC in this second validation cohort.”
- Discussion (page 16): “Even though these specific results should be interpreted with caution considering the limited sample size, the positive association of CSF DDC with UPDRS-III, α -syn load in different brain areas, α -syn Braak stage and DLB stages supports the potential of this marker to track disease progression already from very early stages.”

The cohort descriptions are a bit difficult to follow about DaT Scans, AD CSF signatures etc. The main marker, that could have supported the clinical diagnosis well would have been this new synuclein RT-Quic.

In table 1 it is obvious, that the controls were youngest in all 4 cohorts.

The controls of the “autopsy cohort” were not autopsied and the number and name as autopsy cohort is therefore not correct.

Considering also the comments from the first reviewer we have now adapted the method section and we expect that it can be read more easily now. As is often the case for CSF biomarker studies, controls are younger to the disease group and therefore analyses are performed including age as covariate. Considering that none of controls are autopsy confirmed we have now changed the nomenclature to DLB/AD autopsy cohort and make clear that controls are not autopsy confirmed. Corresponding figures have also been adapted. In line with the comments from the second reviewer, specific abbreviations have also been changed to autopsy confirmed DLB and AD (aDLB and aAD respectively).

Also, it would have been good to see the independent validation of the identified panel by other assays (e.g., ELISA or MS) and not by the same platform that is still an antibody assay, that may identify changes upon the antibody combination. The antibody combination should be written.

In line with reviewer’s comment and as part of our follow-up studies we have also developed an in-house DDC immunoassay and observed again that CSF DDC was increased in DLB compared with controls and AD (see figure below; Bolsewig et al, in preparation), supporting the robustness of the DLB PEA proteome findings. This information has only been included in the discussion (and thus not in the results, as it is part of our ongoing analysis) as follow: “Of note, we have also developed a DDC immunoassay that have shown similar changes in the clinical validation cohort 1, supporting the robustness of the findings (Bolsekig et al, in preparation).”

Moreover, we already validate part of our AD PEA proteome data²⁶ with in-house or commercially available immunoassays^{27,28}, further supporting the validity of the approach.

Reviewers figure 2. CSF DDC levels as measured with an in-house immunoassay.

An in-house DDC immunoassay was developed on automated Ella™ platform using mouse monoclonal and goat polyclonal anti-DDC antibodies for capture and detection, respectively. This assay has been validated according to international guidelines (Andreasson et al, Frontiers Neurology, 2015). We again observed increased levels of CSF DDC in DLB in the same clinical validation cohort 1 used in this study. These analysis also included a group of patients with mild cognitive impairment that were positive for A β pathology (MCI, n= 55), and thus are at the prodromal phase of AD. *** $p < 0.001$. Bolsewig et al, in preparation.

DDC seems to be a marker protein, which is likely changed due to dopaminergic medication in the LBD group. In general, the medication and the UPDRS (as mentioned above) should be given in table 1 and also the levodopa

equivalent dosage. The explanation that even in a subset of 55 DLB subjects, that were not on medication, DDC was increased is limited as dopaminergic medication could also be taken if necessary and not continuously.

In line with reviewer's comments we double checked all the data sets for additional dopaminergic medication. The clinicians confirmed that reports were correct and patients that were not on medication did not take parkinsonian related treatment at the time of sample collection. Nevertheless, information related to levodopa equivalent dosage was not available. Please note that there are cases for which this information was not available, but these were not included in the sensitivity analyses. We recovered additional medication information from a subset of DLB cases from discovery cohort (Penn University, 5 without medication and 8 with medication) and performed additional analysis. Sensitivity analyses using only cases from which parkinsonian medication information was known (190 Controls, 73 DLB cases (13 with parkinsonian medication)) showed that CSF FCER2 was decreased only in those DLB cases undergoing parkinsonian medication. Despite medication moderately influenced the abundance of CSF DDC and CRH, these markers were still dysregulated in DLB patients without any parkinsonian treatment, indicating that medication did not drive the main differences on CSF DDC and CRH observed in DLB (supplementary figure 3). Of note, this is further confirmed by the proteome data obtained in the PPMI PD cohort, as these CSF proteins (including CSF FCER2) were dysregulated in prodromal PD cases who did not have undergone parkinsonian medication by the time of sample collection. Importantly, both DDC and the protein panel discriminate DLB cases with no medication from controls and AD with similar performance to that obtained with the complete cohort (supplementary figure 3). Despite the number of cases that used parkinsonian medication was limited (n=13), and additional analyses are needed to confirm the relevance of treatment effect on CSF FCER2 abundance, this data highlights the importance of including treatment information in biomarker studies. This information has now been included in the supplementary figure 3 and in the main text as follows:

- Results, page 11: *"Sensitivity analysis using only cases from which parkinsonian medication information was known (190 Controls (no medication), 73 DLB cases (only 13 had parkinsonian medication)) showed that CSF FCER2 was decreased only in those DLB cases undergoing treatment (supplementary figure 3a). Despite medication moderately influenced the abundance of CSF DDC and CRH, these markers were still dysregulated in DLB patients without any parkinsonian treatment (supplementary figure 3a). Noteworthy, both DDC and the protein panel discriminate DLB cases with no medication from controls and AD with similar performance to that obtained with the complete cohort (supplementary figure 3b,c)".*
- Results, page 13: *"Please note that prodromal PD do not undergo dopaminergic medication, further supporting that the changes observed between groups are not driven by parkinsonian related treatment."*
- Discussion, page 16: *"We observed that parkinsonian medication moderately influenced the levels of some CSF proteins. Despite the number of cases that underwent parkinsonian medication was limited (n=13), such data highlights the importance of including treatment information on biomarker studies. Additional analyses are needed to confirm the relevance of treatment effect on CSF FCER2 abundance. Importantly, medication did not drive the main differences observed between groups for the other DLB related proteins (i.e., CSF, CRH)".*

Minor:

Often in the manuscript the direction is not given and writing that there is a change in this and that marker does lack this information on the direction.

We have now specified the direction of the changes along the manuscript when referring to specific markers.

There should be more discussion on the pathophysiology of the other markers than DDC and possibly some pathway analysis.

In line with reviewers comments we have now further elaborated the pathophysiology of other proteins and pathways (page 14):

“Interestingly, the three top proteins dysregulated in DLB (i.e., DDC, CRH and GH) can regulate the hypothalamic-pituitary-adrenal (HPA) axis^{29,30}, suggesting that one of the major systems associated to stress response and behavioral dysfunction might be involved in DLB pathophysiology. This is in line with previous studies showing hypothalamic dysfunction in DLB^{31,32}.”

Please note that two additional co-authors have been included (Juliette L. van Alphen and Sanaz Arezoumandan), who have been involved in the collection of the new data.

References

1. Goetz, C. G. *et al.* Movement Disorder Society-sponsored revision of the Unified Parkinson's Disease Rating Scale (MDS-UPDRS): Scale presentation and clinimetric testing results. *Movement Disorders* **23**, 2129–2170 (2008).
2. McKeith, I. G. *et al.* Diagnosis and management of dementia with Lewy bodies: Fourth consensus report of the DLB Consortium. *Neurology* **89**, 88–100 (2017).
3. van de Beek, M. *et al.* Characterization of symptoms and determinants of disease burden in dementia with Lewy bodies: DEvelop design and baseline results. *Alzheimers Res Ther* **13**, (2021).
4. van der Flier, W. M. & Scheltens, P. Amsterdam Dementia Cohort: Performing Research to Optimize Care. *Journal of Alzheimer's Disease* **62**, 1091–1111 (2018).
5. Alcolea, D. *et al.* The Sant Pau Initiative on Neurodegeneration (SPIN) cohort: A data set for biomarker discovery and validation in neurodegenerative disorders. *Alzheimer's and Dementia: Translational Research and Clinical Interventions* **5**, 597–609 (2019).
6. Cummings, J. L. The Neuropsychiatric Inventory. *Neurology* **48**, 10S-16S (1997).
7. Eikelboom, W. S. *et al.* Neuropsychiatric and Cognitive Symptoms Across the Alzheimer Disease Clinical Spectrum: Cross-sectional and Longitudinal Associations. *Neurology* **97**, e1276 (2021).
8. Teunissen, C. E. *et al.* Blood-based biomarkers for Alzheimer's disease: towards clinical implementation. *Lancet Neurol* (2021) doi:10.1016/S1474-4422(21)00361-6.
9. Feleke, R. *et al.* Cross-platform transcriptional profiling identifies common and distinct molecular pathologies in Lewy body diseases. *Acta Neuropathologica* **2021** *142*:3 **142**, 449–474 (2021).
10. Pietrzak, M. *et al.* Gene expression profiling of brain samples from patients with Lewy body dementia. *Biochem Biophys Res Commun* **479**, 875 (2016).
11. Sjöstedt, E. *et al.* An atlas of the protein-coding genes in the human, pig, and mouse brain. *Science* (1979) **367**, (2020).
12. Vogel, C. & Marcotte, E. M. Insights into the regulation of protein abundance from proteomic and transcriptomic analyses. *Nature Reviews Genetics* **2012** *13*:4 **13**, 227–232 (2012).
13. Van Steenoven, I. *et al.* Identification of novel cerebrospinal fluid biomarker candidates for dementia with Lewy bodies: a proteomic approach. *Mol Neurodegener* **15**, 36 (2020).
14. Abdi, F. *et al.* Detection of biomarkers with a multiplex quantitative proteomic platform in cerebrospinal fluid of patients with neurodegenerative disorders. *J Alzheimers Dis* **9**, 293–348 (2006).
15. Dieks, J. K. *et al.* Low-abundant cerebrospinal fluid proteome alterations in dementia with Lewy bodies. *J Alzheimers Dis* **34**, 387–397 (2013).
16. Wang, Z.-Y. *et al.* Use of CSF α -synuclein in the differential diagnosis between Alzheimer's disease and other neurodegenerative disorders. *Int Psychogeriatr* (2015) doi:10.1017/S1041610215000447.
17. Slaets, S. *et al.* Increased CSF α -synuclein levels in Alzheimer's disease: Correlation with tau levels. *Alzheimer's & Dementia* (2014) doi:10.1016/j.jalz.2013.10.004.
18. Kasuga, K., Nishizawa, M. & Ikeuchi, T. α -synuclein as CSF and blood biomarker of dementia with lewy bodies. *International Journal of Alzheimer's Disease* vol. 2012 Preprint at <https://doi.org/10.1155/2012/437025> (2012).
19. Groveman, B. R. *et al.* Rapid and ultra-sensitive quantitation of disease-associated α -synuclein seeds in brain and cerebrospinal fluid by α Syn RT-QuIC. *Acta Neuropathol Commun* **6**, 7 (2018).
20. Rossi, M. *et al.* Ultrasensitive RT-QuIC assay with high sensitivity and specificity for Lewy body-associated synucleinopathies. *Acta Neuropathol* **140**, 49–62 (2020).
21. Wang, Z. *et al.* Skin α -Synuclein Aggregation Seeding Activity as a Novel Biomarker for Parkinson Disease. *JAMA Neurol* **78**, 30–40 (2021).
22. Rossi, M. *et al.* Diagnostic Value of the CSF α -Synuclein Real-Time Quaking-Induced Conversion Assay at the Prodromal MCI Stage of Dementia With Lewy Bodies. *Neurology* [10.1212/WNL.00000000000012438](https://doi.org/10.1212/WNL.00000000000012438) (2021) doi:10.1212/WNL.00000000000012438.

23. Fairfoul, G. *et al.* Alpha-synuclein RT-QulC in the CSF of patients with alpha-synucleinopathies. *Ann Clin Transl Neurol* **3**, 812–818 (2016).
24. Bellomo, G. *et al.* Cerebrospinal fluid lipoproteins inhibit α -synuclein aggregation by interacting with oligomeric species in seed amplification assays. (2023).
25. Walker, L., Stefanis, L. & Attems, J. Clinical and neuropathological differences between Parkinson's disease, Parkinson's disease dementia and dementia with Lewy bodies – current issues and future directions. *Journal of Neurochemistry* vol. 150 467–474 Preprint at <https://doi.org/10.1111/jnc.14698> (2019).
26. del Campo, M. *et al.* CSF proteome profiling across the Alzheimer's disease spectrum reflects the multifactorial nature of the disease and identifies specific biomarker panels. *Nat Aging* (2022) doi:10.1038/s43587-022-00300-1.
27. Hok-A-Hin, Y. S. *et al.* Neuroinflammatory CSF biomarkers MIF, sTREM1, and sTREM2 show dynamic expression profiles in Alzheimer's disease. *J Neuroinflammation* **20**, (2023).
28. Yanaika S. Hok-A-Hin *et al.* Thimet Oligopeptidase as a potential CSF biomarker for Alzheimer's Disease. A cross-platform validation study. *Alzheimer's & Dementia: Diagnosis, Assessment & Disease Monitoring* **in press**, (2023).
29. Belda, X. & Armario, A. Dopamine D1 and D2 dopamine receptors regulate immobilization stress-induced activation of the hypothalamus-pituitary-adrenal axis. *Psychopharmacology (Berl)* **206**, 355–365 (2009).
30. Tsigos, C. & Chrousos, G. P. Hypothalamic–pituitary–adrenal axis, neuroendocrine factors and stress. *J Psychosom Res* **53**, 865–871 (2002).
31. Kristofer Schultz *et al.* Reduced CSF CART in dementia with Lewy bodies. *Neurosci Lett* (2009) doi:10.1016/j.neulet.2009.02.008.
32. Whitwell, J. L. *et al.* Focal atrophy in Dementia with Lewy Bodies on MRI: a distinct pattern from Alzheimer's disease NIH Public Access. *Brain* (2007) doi:10.1093/brain/awl388.

REVIEWERS' COMMENTS

Reviewer #1 (Remarks to the Author):

I saw that the author's revisions improved the manuscript significantly. I have no further comments.

Reviewer #3 (Remarks to the Author):

The group has done extensive effort to analyze the heterogeneity of the cohorts by adding other symptoms and DaT if available. My major concern is that the authors have not described how they evaluated RBD in these cohorts, which is necessary as the state of the art is still polysomnography.

I would also encourage the group to rather use the more commonly used term Seed Amplification Assay than RT-QuIC throughout the manuscript.

The sentence: "RT-QuIC is a relatively novel technology that

have shown highly promising findings^{19,20,23}. Still, it is laborious and requires further optimization before its

implementation in research and clinical settings²⁴."

I think needs revision as the company Amprion to my knowledge offers their assay from a Clia lab as routine.

I appreciate the new nomenclature of autopsy confirmed cases and the inclusion of at least some PD data from an available cohort.

Reviewer #1 (Remarks to the Author):

I saw that the author's revisions improved the manuscript significantly. I have no further comments.

We thank the reviewer for the positive comment.

Reviewer #3 (Remarks to the Author):

The group has done extensive effort to analyze the heterogeneity of the cohorts by adding other symptoms and DaT if available. My major concern is that the authors have not described how they evaluated RBD in these cohorts, which is necessary as the state of the art is still polysomnography.

We thank the reviewer for the positive comment.

We have now specified the evaluation of RBD as follows (page 13): *“Evaluation of REM Sleep Behavior Disorder (RBD) includes an interview with a sleep specialist, a full nocturnal video-polysomnography, the Mayo Sleep questionnaire (cutoff ≥ 1) or caregivers reporting that a patient would ‘act out’ their dreams and moves extensively during their sleep as previously described¹⁻³”*

I would also encourage the group to rather use the more commonly used term Seed Amplification Assay than RT-QuIC throughout the manuscript.

In agreement with reviewer’s comment, RT-QuIC have been changed for seed amplification assays throughout the manuscript.

The sentence: “RT-QuIC is a relatively novel technology that have shown highly promising findings^{19,20,23}. Still, it is laborious and requires further optimization before its implementation in research and clinical settings²⁴.” I think needs revision as the company Amprion to my knowledge offers their assay from a Clia lab as routine.

We agree with reviewer's comment. This sentence was included however only in the previous reviewers' response to explain why α -syn pathology was not measured using seed amplification assays across the different cohorts. No additional changes have thus been done within the revised manuscript.

I appreciate the new nomenclature of autopsy confirmed cases and the inclusion of at least some PD data from an available cohort.

We appreciate reviewer's comment.

Reference

1. Alcolea, D. *et al.* The Sant Pau Initiative on Neurodegeneration (SPIN) cohort: A data set for biomarker discovery and validation in neurodegenerative disorders. *Alzheimer's and Dementia: Translational Research and Clinical Interventions* **5**, 597–609 (2019).
2. van de Beek, M. *et al.* Association of the ATN Research Framework With Clinical Profile, Cognitive Decline, and Mortality in Patients With Dementia With Lewy Bodies. *Neurology* 10.1212/WNL.000000000200048 (2022) doi:10.1212/wnl.000000000200048.
3. van de Beek, M. *et al.* Characterization of symptoms and determinants of disease burden in dementia with Lewy bodies: DEvELOP design and baseline results. *Alzheimers Res Ther* **13**, (2021).